# BEYOND WORDS: MULTIMODAL LLM KNOWS WHEN TO SPEAK

## ABSTRACT

While large language model (LLM)-based chatbots have demonstrated strong capabilities in generating coherent and contextually relevant responses, they often struggle with understanding when to speak, particularly in delivering brief, timely reactions during ongoing conversations. This limitation arises largely from their reliance on inputs of limited modality, lacking the rich contextual cues in real-world human dialogue. In this work, we focus on real-time prediction of response types, with an emphasis on short, reactive utterances that depend on subtle, multimodal signals across video, speech, and text. To support this, we introduce a new multimodal dataset constructed from real-world conversational videos, containing temporally aligned visual, auditory, and textual streams. This dataset enables fine-grained modeling of response timing in dyadic interactions. Building on this dataset, we propose MM-When2Speak, a multimodal LLM-based model that adaptively integrates visual, auditory, and textual context to predict when a response should occur, and what type of response is appropriate. Experiments show that MM-When2Speak outperforms state-of-the-art LLM-based baselines, achieving up to a 4× improvement in response timing accuracy over leading commercial LLMs. These results underscore the importance of multimodal inputs for producing timely, natural, and engaging conversational AI. We will release code and dataset to support further research.

## 1 INTRODUCTION

Recently, many AI-based systems, particularly digital chatbots, have begun to rely on more intelligent interactions with humans, engaging in conversations that align with human speech patterns. Beyond generating accurate response content, these agents are increasingly expected to know when to respond, not just what to say. In real human conversations, individuals alternate between speaking and listening roles based on subtle contextual cues, social norms, and speaker intention (Wang et al., 2024a; Ekstedt & Skantze, 2020; Stivers et al., 2009; Umair et al., 2024b). Moreover, listeners frequently provide brief verbal signals (*backchannel reactions*) that convey attention, agreement, or understanding without disruption (Wittenburg et al., 2006; Yngve, 1970). The exact predictions of the moments for these responses and reactions to take place is crucial for ensuring a fully interactive and engaging human-machine conversations.

Many state-of-the-art large language models (LLMs), such as Achiam et al. (2023); Liu et al. (2024); Ekstedt & Skantze (2020); Ekstedt et al. (2023), are designed to engage with humans in a structured, turn-based manner. However, they often lack the ability to predict appropriate timing for backchannel responses, which is important for fostering social engagement, and consequently struggle to capture the fluidity and dynamism of natural human dialogue (Wang et al., 2024a). Some prior work has directly addressed the task of backchannel detection (Amer et al., 2023; Wang et al., 2023; Park et al., 2024; Inoue et al., 2025; Fukunaga et al., 2025; Cieri et al., 2004; Bilakhia et al., 2015; Wu et al., 2020; Lee et al., 2025). However, these methods often overlook the diversity and contextual variability of backchannel words, limiting their ability to support nuanced and naturalistic conversational behavior.

Enabling such responsiveness remains highly challenging, especially in determining when to speak and when to react with backchannel cues. Progress is hindered by the limited integration of multimodal information needed to interpret conversational dynamics. Specifically, two key challenges arise: **(1) On the dataset side**, most existing resources rely on written corpora (Umair et al., 2024b;

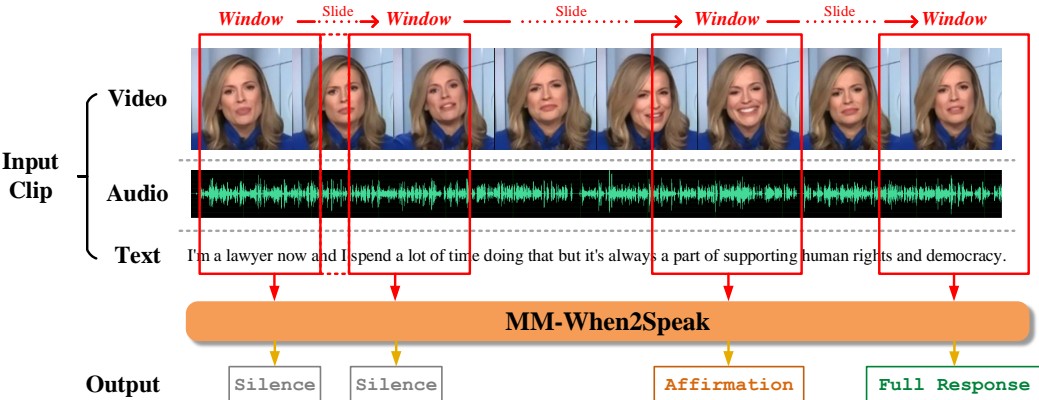

Figure 1: Overview of MM-When2Speak. It uses a sliding window to densely sample short clips for response type prediction, transforming "when to speak" problem to a classification task. At each sampled timestamp, it outputs a specific label, indicating whether to keep silent, give a short reaction (e.g., `Affirmation`), or start responding.

Pilán et al., 2024; Mahowald et al., 2024; Cieri et al., 2004), or at best audio-auxiliary datasets (Bilakhia et al., 2015; Lee et al., 2025), which substantially underrepresent the role of visual signals. Yet spoken interactions are inherently fragmented, fast-paced, and context-dependent, detecting turn boundaries or recognizing subtle opportunities for backchannel reactions often requires access to multimodal cues, such as gaze, head nods, and facial expressions, which text or audio alone cannot adequately capture. **(2) On the modeling side**, many LLM-based systems follow a text-centric pipeline: speech is first transcribed via automatic speech recognition (ASR), then processed by an LLM, and subsequently converted back into audio through text-to-speech (TTS). While being effective for semantic processing, this pipeline tends to disregard crucial auditory cues (e.g., prosody) and visual cues (e.g., facial expressions) that humans naturally exploit to anticipate turn transitions and produce timely reactions. As a result, current systems often lack the contextual grounding necessary for real-time, socially appropriate responsiveness, which limits their ability to support fluid, engaging, and human-like interactions.

To address these challenges, we introduce a multimodal approach for conversational agents to predict "when to speak and react" in a manner that aligns with the dynamics of real-world human interaction. We begin by constructing a carefully curated dataset consisting of synchronized visual (talking-head video), auditory (speech), and textual (transcriptions) data drawn from dyadic conversations. We define two basic action categories: `full_response`, indicating that the listener takes the speaking turn, and `silence`, indicating no voice from the listener at all. To capture the nuanced cues that precede short, context-sensitive listener reactions, we further annotate the dataset with seven distinct reaction types, each representing a brief but meaningful form of listener feedback: `affirmation`, `gratitude`, `farewell`, `greeting`, `question`, `surprise`, and `pondering`. These categories enable a more fine-grained modeling of conversational timing by linking diverse communicative intents to subtle multimodal signals. In doing so, our approach moves beyond coarse turn-taking paradigms and toward a richer understanding of "when to speak", especially the fine-grained modeling of "when to react", in natural, real-time dialogue.

Building on these insights, we propose MM-When2Speak, a multimodal LLM-based architecture, as shown in Fig. 1, which jointly predicts when to respond, to react, or to keep silent. The model adaptively integrates visual, auditory, and textual information, framing the task as a multi-class classification problem. Furthermore, a sliding-window mechanism enables online inference, supporting real-time responsiveness. This unified framework can enable the system to generate both full responses and brief, in-turn reactions grounded in multimodal context. We evaluate our approach against a suite of state-of-the-art LLMs on our newly collected dataset , and experimental results show that our method significantly outperforms these baselines across all key metrics. Notably, MM-When2Speak achieves up to a 4× improvement in response type prediction performance, demonstrating the crucial role of multimodal input in further enabling accurate and natural conversational behavior.

## 2 RELATED WORK

### 2.1 RESPONSE TIMING AND BACKCHANNEL DETECTION

Speaking turn prediction and backchannel detection has been widely studied to mimic real-world human-like conversation Umair et al. (2024b). Traditional dialogue systems rely on fixed silence thresholds for turn shifts, often resulting in delayed or overlapping turns (Umair et al., 2024b; Wang et al., 2024a), while modern transformer-based models have been applied to predict *transition relevance places* and backchannel opportunities. For instance, Ekstedt & Skantze (2020) introduces TurnGPT, fine-tuning GPT-2 on turn-taking data and outperforming prosody-only baselines in anticipating turn ends.Amer et al. (2023) adopts a simple transformer architecture for automatic backchannel analysis based on pose and facial information. Umair et al. (2024a) demonstrates that modern LLMs "know what to say but not when to speak", missing within-utterance openings and advocating new Transition Relevance Places (TRPs) benchmarks. Wang et al. (2024b) proposes a full-duplex LLM-based speech dialogue scheme where the model emits control tokens to decide between listening, interrupting, or responding, achieving more precise interrupting behaviors than commercial assistants. Fukunaga et al. (2025), based on a Japanese dataset, designs a multimodal joint-listener-speaker architecture for a 3-class backchannel classification task. (Müller et al., 2021; 2022; 2023; 2024) releases a public dataset, namely MPIIGroupInteraction, for multi-participant conversational engagement estimation, facilitating the studies of turn-taking and backchannel detection. In this paper, we refine the backchannel detection task jointly with response timing prediction with the proposed MM-When2Speak in dyadic conversational scenarios.

### 2.2 MULTIMODAL LARGE LANGUAGE MODELS

With the integration of multimodal input, large language models transcend their previous capability boundaries, exhibiting statistically significant performance gains across diverse application domains and evaluation benchmarks. Alayrac & et al. (2022) fuses a frozen LLM with visual encoders for few-shot image understanding, matching state-of-the-art in open-ended VQA without additional fine-tuning. In addition, Li et al. (2023) and Zhu & et al. (2023) introduce lightweight visual adapters to enable conversational image description and reasoning. Extending these ideas to video, Maaz et al. (2024) trains on video–instruction datasets for fluent temporal dialogue about dynamic scenes. For end-to-end speech understanding and generation, Bhandari & et al. (2023) unifies PaLM-2 (Anil et al., 2023) with AudioLM (Borsos et al., 2023), achieving SOTA in speech translation while preserving speaker voice and prosody. Furthermore, Driess & et al. (2023) interleaves continuous robot sensor streams with text, enabling embodied reasoning and manipulation planning within a single LLM context. More recently, Wang et al. (2024d) introduces a "video–text duet" format, allowing the model to interject in real time during video playback, significantly improving dense captioning and temporal grounding tasks. In this paper, we leverage the multimodal potential to equip LLMs with the ability to predict "when to speak" during human conversations.

## 3 METHODOLOGY

### 3.1 PROBLEM FORMULATION

We consider a conversation between a *user* and a *machine*, where the machine is responsible for determining whether it should produce a brief reaction, take the conversational turn to deliver a full response, or remain silent. We formulate this decision-making process as a dense classification task over short conversational clips, where each clip is assigned one of the following response type labels: $\hat{y} = \{$affirmation, gratitude, farewell, greeting, question, surprise, pondering, full_response, silence$\}$. The first seven labels correspond to brief listener reactions, which are minimal verbal cues that signal engagement or emotion without taking the conversational floor. The label full_response indicates a turn-shifting action, where the machine is expected to take over and respond substantively. The label silence denotes that the machine should remain quiet.

To enable fine-grained, real-time predictions from continuous input, we adopt a sliding window approach over the full video. Specifically, the user's input is segmented into overlapping clips using a fixed window size of $\Delta t = 10$ seconds and a stride of $\delta = 0.5$ seconds, as illustrated in Fig. 1. Let the full video be denoted as $\mathcal{V} = (V_{\mathcal{V}}, S_{\mathcal{V}}, T_{\mathcal{V}})$, where $V_{\mathcal{V}} \in \mathbb{R}^{f_{\mathcal{V}}^v \times h \times w \times 3}$ represents the sequence

of RGB video frames of spatial resolution $h \times w$ with $f_{\mathcal{V}}^v$ total frames, $S_{\mathcal{V}} \in \mathbb{R}^{f_{\mathcal{V}}^s \times d}$ is the audio signal with $f_{\mathcal{V}}^s$ frames and feature dimension $d$, and $T_{\mathcal{V}} = (w_1, w_2, \ldots, w_{n_{\mathcal{V}}})$ is the transcribed user dialogue containing $n_{\mathcal{V}}$ tokens. Each clip $C_i$ is extracted with end time $t_i = \Delta t + (i-1)\delta$ as $C_i = (V_{C_i}, S_{C_i}, T_{C_i}) = \mathrm{Crop}(\mathcal{V}, t_i - \Delta t, t_i)$, where $V_{C_i} \in \mathbb{R}^{f_{\Delta t}^v \times h \times w \times 3}$, $S_{C_i} \in \mathbb{R}^{f_{\Delta t}^s \times d}$, and $T_{C_i} = (w_1, \ldots, w_{n_{\Delta t}})$, with $f_{\Delta t}^v$, $f_{\Delta t}^s$, and $n_{\Delta t}$ denoting the number of video frames, audio frames, and transcript tokens in each $\Delta t$-second window.

The machine processes each clip independently and predicts a corresponding response type, resulting in an output sequence $\hat{\boldsymbol{y}} = \mathrm{Cls}(\mathcal{C})$, where $\mathcal{C} = \{C_i\}_{i=1}^N$ is the set of $N$ clips sampled from the full video $\mathcal{V}$ and $\hat{\boldsymbol{y}} = \{\hat{y}_i\}_{i=1}^N$ is the set of predicted response type labels, one for each clip. These clips are densely and uniformly extracted to enable real-time prediction of when the machine should speak, with high temporal resolution allowing precise alignment with conversational dynamics. For each predicted label $\hat{y}_i$ at the end of clip $C_i$, which concludes at time $t_i$: (1) if $\hat{y}_i = \texttt{silence}$, the machine remains quiet; (2) if $\hat{y}_i = \texttt{full\_response}$, the machine initiates a turn and begins speaking at $t_i$; (3) otherwise, the machine produces a brief reaction of the specified type at $t_i$ of the associated reaction types without taking the conversational turn.

## 3.2 Collection and Curation of Our Dataset

We collect over 2,000 dyadic conversation videos from public online platforms such as YouTube, each featuring two individuals engaged in frontal-face, split-screen dialogues in either virtual Zoom meetings or broadcast news segments (e.g., CNN interviews). To ensure data quality, we apply a series of filtering steps to remove low-quality samples, excluding videos with poor or inconsistent face visibility, excessive background noise unrelated to the conversation, or substantial overlapping speech that hinders reliable transcription. After this pre-processing pipeline, we retain 377 high-quality, full-length videos for further annotation.

Each video is first segmented into clips/segments as described in Sec. 3.1. We then annotate the response types of these clips via audio diarization. Specifically, if a speaker begins speaking while the other ceases, it is marked as a $\texttt{full\_response}$, indicating a turn switch; if a speaker produces a brief utterance at a time without interrupting the other speaker, the clip is labeled as a $\texttt{reaction}$; otherwise, the clip is labeled as $\texttt{silence}$. To further classify the type of each $\texttt{reaction}$ clip, we provide its text transcript to ChatGPT and categorize the responses into seven fine-grained reaction types as described in Sec. 3.1.

From the labeled clips collected from 357 original videos, we construct the **Short-Clips** dataset by randomly sampling 4,393 $\texttt{reaction}$, 2,000 $\texttt{full\_response}$, and 2,000 $\texttt{silence}$ segments. This dataset is divided into training and testing subsets by a ratio of 7:3, denoted as **Short-Clips-Train** for model training and **Short-Clips-Test** for evaluation, respectively. The dataset is specifically designed to assess a model's ability to predict appropriate vocal responses from isolated short clips, independent of broader conversational context.

In addition, we also construct the **Full-Videos** dataset using the remaining 20 videos for evaluating the model's ability to determine "when to speak" in a continuous dialogue setting. For each video in this dataset, we retain all overlapping clips with a window size of 10 seconds and a stride of 0.5 seconds, resulting in a dense sequence of time-aligned response-type labels that reflect the model's speaking and reaction decisions. This setup enables fine-grained temporal evaluations. We also introduce an extra parameter $\tau$ to handle uncertainty and ambiguity of overlapping, which is explained in the appendix Sec. A.3.2.

To verify the annotation quality, we conduct a human evaluation study on the Short-Clips dataset, as described in Sec. 4.4 below. The results show that 92 out of 105 samples (87.62%) are correctly categorized in alignment with human judgments, indicating high annotation reliability. Additional details on the data collection pipeline, segment pre-processing, response type annotation, dataset construction, and uncertainty handling are provided in the appendix Sec. A.

## 3.3 MM-When2Speak

We design our MM-When2Speak model to integrate visual, audio, and textual modalities for jointly classifying input clips and identifying appropriate speaking times within a conversation. We adopt an "Encoder-Adaptor-LLM" pipeline similar to Fu et al. (2025), combining vision and audio transformers

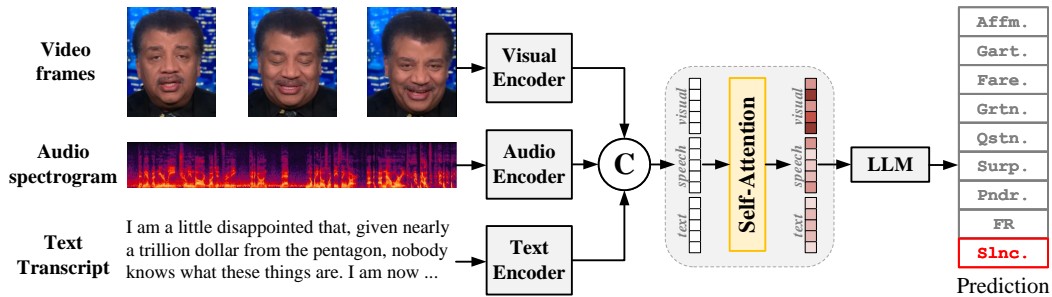

Figure 2: Architecture of our MM-When2Speak. It encodes videos frames, spectrogram features and tokenized texts for multimodal information perception, and adaptively combines these attentive information for the LLM to accurately identify the correct response type prediction[1].

as well as MLP adaptors with an LLM. For the vision modality, we adopt InterViT Chen et al. (2024) as the visual encoder, outputting 256 tokens each frame; for audio modality, we use Mel filters to convert the raw audio data, and further use 4 convolutional layers for 4x downsampling and 24 Transformer blocks to encode audio features following Wang et al. (2024c); for text modality, we choose Qwen2-7B (Yang et al., 2024a) as our LLM. We fix LLM input embedding dimension to 4,096.

**Multimodal fusion.** We adopt a simple yet effective multimodal fusion strategy that selectively emphasizes salient features across modalities to improve fine-grained prediction. Concretely, we apply a self-attention mechanism over concatenated embeddings, allowing the model to adaptively prioritize modality-specific signals most relevant to the task and remain robust under modality imbalance. A high-level overview is shown in Fig. 2, with details provided in the appendix Sec. B.

**Training strategy.** We adopt a two-stage training strategy inspired by Fu et al. (2025), initially pretraining our model to encourage cross-modal consistency among audio, visual, and textual representations. This pretraining phase aligns feature distributions across modalities, allowing the model to develop robust multimodal embeddings. Following pretraining, we perform full supervision fine-tuning on our Short-Clips-Train dataset to adapt the model for response classification. Fine-tuning is conducted using two NVIDIA L40 GPUs with a batch size of 32, a learning rate of 1e-5, and the AdamW optimizer. To address class imbalance, particularly the over-representation of certain reaction classes, we apply Focal Loss with a focusing parameter gamma as 2 and a weighting factor alpha as 0.5. This loss formulation reduces the relative loss contribution from well-classified examples, placing more emphasis on harder, underrepresented instances.

## 4 EXPERIMENTS

### 4.1 EXPERIMENTAL SETUP

We evaluate the response-type prediction performance of MM-When2Speak against a range of state-of-the-art models under various modality configurations in the zero-shot setting. Task-specific prompts are designed to constrain model outputs to predefined response labels. The full prompt template is included in appendix Sec. C. For backchannel detection task specifically, we compare MM-When2Speak to a baseline method tailored for backchannel detection in appendix Sec. G; implementation and configuration details are also provided in appendix Sec. A and B.

In the Text setting, we include ChatGPT-4o, DeepSeek-V3:7b, Qwen-2.5:7b, Mistral:8b, Gemma2:9b, and VITA-1.5; in the Speech+Text setting, ChatGPT-4o, Qwen2-Audio, Gemini-1.5, and VITA-1.5; in the Video+Text setting, ChatGPT-4o, Gemini-1.5, Qwen2.5-VL, and VITA-1.5; and in the full Video+Speech+Text setting, we compare against VITA-1.5, one of the few public models supporting all three modalities. For proprietary models (ChatGPT-4o, Gemini-1.5), inference is conducted via official APIs, while other models are deployed locally on a single NVIDIA L40s GPU.

---

[1]Affm.= affirmation, Grat.= gratitude, Fare.= farewell, Grtn.= greeting, Qstn.= question, Surp.= surprise, Pndr.= pondering, FR.= full_response, Slnc.= silence.

Table 1: Performance evaluations of response type prediction on Short-Clips-Test dataset for methods of different modalities. P: precision, R: recall.

| Method | Metric | Affm. | Grat. | Fare. | Grtn. | Qstn. | Surp. | Pndr. | FR | Slnc. |
|---|---|---|---|---|---|---|---|---|---|---|
| Text only | | | | | | | | | | |
| ChatGPT-4o | P | 18.30 | 15.96 | 17.51 | 16.67 | 14.67 | 14.06 | 13.92 | 18.68 | 23.21 |
| | R | 15.02 | 12.72 | 14.25 | 13.42 | 11.46 | 10.87 | 10.73 | 15.40 | 19.87 |
| DeepSeek-V3 | P | 13.74 | 19.90 | 20.63 | 22.39 | 12.85 | 12.26 | 12.32 | 20.25 | 21.91 |
| | R | 10.56 | 16.60 | 17.32 | 19.06 | 9.69 | 9.12 | 9.18 | 16.94 | 18.58 |
| Qwen-2.5 | P | 19.74 | 16.76 | 14.37 | 13.05 | 13.03 | 14.39 | 14.39 | 20.64 | 21.88 |
| | R | 16.44 | 13.51 | 11.17 | 9.89 | 9.87 | 7.68 | 11.19 | 17.33 | 18.55 |
| Mistral | P | 18.45 | 17.00 | 13.55 | 16.13 | 11.81 | 14.94 | 14.21 | 20.73 | 24.48 |
| | R | 15.17 | 13.74 | 10.37 | 12.89 | 8.69 | 11.73 | 11.01 | 17.42 | 21.13 |
| Gemma2 | P | 19.22 | 18.97 | 22.45 | 15.34 | 11.67 | 12.35 | 14.90 | 19.48 | 22.94 |
| | R | 15.93 | 15.68 | 19.12 | 12.12 | 8.55 | 11.69 | 16.18 | 19.60 | |
| VITA-1.5 | P | 15.32 | 17.08 | 15.19 | 14.04 | 14.78 | 12.68 | 13.34 | 20.46 | 21.49 |
| | R | 12.10 | 13.82 | 11.97 | 10.85 | 11.57 | 9.53 | 10.17 | 17.15 | 18.17 |
| Speech + Text | | | | | | | | | | |
| ChatGPT-4o | P | 26.85 | 20.76 | 27.76 | 23.68 | 21.07 | 21.57 | 20.70 | 27.05 | 31.43 |
| | R | 24.19 | 18.14 | 25.09 | 21.04 | 18.44 | 18.94 | 18.08 | 24.39 | 28.75 |
| Gemini-1.5 | P | 30.49 | 30.38 | 26.24 | 24.25 | 18.06 | 19.21 | 19.30 | 28.08 | 28.84 |
| | R | 27.81 | 27.70 | 23.58 | 21.60 | 15.46 | 16.60 | 16.69 | 25.41 | 26.17 |
| Qwen2-Audio | P | 29.49 | 24.83 | 21.09 | 19.02 | 18.76 | 19.00 | 15.43 | 25.13 | 30.22 |
| | R | 26.82 | 22.18 | 18.46 | 16.41 | 16.15 | 16.39 | 12.86 | 22.48 | 27.54 |
| VITA-1.5 | P | 24.64 | 20.34 | 26.03 | 25.59 | 16.86 | 24.24 | 16.97 | 30.70 | 32.69 |
| | R | 21.99 | 17.72 | 23.37 | 22.94 | 14.27 | 21.59 | 14.38 | 28.02 | 30.00 |
| Video + Text | | | | | | | | | | |
| ChatGPT-4o | P | 26.96 | 23.89 | 27.31 | 28.95 | 23.30 | 23.83 | 21.85 | 27.49 | 32.05 |
| | R | 25.03 | 21.97 | 25.38 | 27.02 | 21.38 | 21.91 | 19.94 | 25.56 | 30.11 |
| Gemini-1.5 | P | 29.62 | 31.05 | 27.77 | 24.97 | 20.63 | 24.01 | 22.25 | 25.88 | 30.06 |
| | R | 27.69 | 29.11 | 25.84 | 23.05 | 18.72 | 20.34 | 22.09 | 23.95 | 28.12 |
| Qwen2.5-VL | P | 21.89 | 23.87 | 20.32 | 19.69 | 19.30 | 20.88 | 21.19 | 23.78 | 24.52 |
| | R | 17.65 | 19.83 | 17.25 | 17.14 | 16.01 | 15.42 | 16.98 | 20.09 | 19.49 |
| VITA-1.5 | P | 25.04 | 24.23 | 20.88 | 26.81 | 20.69 | 25.09 | 18.56 | 33.17 | 34.96 |
| | R | 23.12 | 22.31 | 18.97 | 24.88 | 18.78 | 23.17 | 16.66 | 31.23 | 33.02 |
| Video + Speech + Text | | | | | | | | | | |
| VITA-1.5 | P | 38.03 | 39.68 | 39.16 | 38.81 | 28.46 | 31.23 | 23.32 | 31.36 | 35.32 |
| | R | 35.14 | 36.79 | 36.27 | 35.92 | 25.61 | 28.37 | 20.50 | 28.50 | 32.44 |
| MM-When2Speak | P | **62.21** | **64.35** | **63.15** | **63.29** | **46.26** | **50.52** | **37.78** | **68.15** | **68.78** |
| | R | **59.86** | **61.99** | **60.79** | **60.44** | **43.91** | **46.25** | **35.45** | **65.79** | **66.42** |

We use precision (P), recall (R), and F1 score metrics to evaluate the response-type prediction performance of each method.

## 4.2 COMPARISONS WITH SOTA LLMs

We conduct comparative experiments on both of our datasets. The Short-Clips-Test dataset emphasizes the ability to accurately classify response-type labels from brief multimodal snippets, whereas the Full-Videos dataset evaluates model performance in more natural real-world conversation scenarios.

Tables 1 and 2 summarize the experimental results. For Short-Clips dataset, as shown in Table 1, model performance improves consistently as additional modalities are incorporated. For instance, ChatGPT-4o with text only yields limited accuracy (e.g., 18.68%/15.40% Precision/Recall for full_response), whereas adding speech and video leads to substantial average gains of 8.31% and 9.24% in Precision and Recall, respectively. A similar pattern is observed for VITA-1.5, which exhibits more than 10% improvements in multiple classes when all three modalities are available. These results confirm that multimodal information is critical for reliable response-type prediction. Moreover, benefiting from our multimodal fusion mechanism based on self-attention, the proposed MM-When2Speak consistently outperforms all baselines, achieving performance up to 4x over ChatGPT-4o (Text), and 1.5x over VITA-1.5 (Video+Speech+Text) on the Short-Clips-Test dataset.

For the Full-Videos dataset, we apply a sliding-window strategy to align with the workflow of the Short-Clips-Test dataset, predicting the response type for each 10-second segment. As reported in Table 2, we observe a consistent trend with Table 1: LLM performance improves with additional modalities, and MM-When2Speak achieves the strongest overall results. Nevertheless, absolute performance is lower than in Table 1, likely due to distributional differences between the two datasets. In particular, the Full-Videos dataset better reflects real-world conversational dynamics, where dense

Table 2: Performance evaluations of response type prediction on Full-Videos dataset for methods of different modalities. P: precision, R: recall.

| Method | Metric | Affm. | Grat. | Fare. | Grtn. | Qstn. | Surp. | Pndr. | FR | Slnc. |
|---|---|---|---|---|---|---|---|---|---|---|
| | | | | | Text only | | | | | |
| ChatGPT-4o | P | 10.85 | 8.83 | 8.09 | 8.17 | 8.05 | 8.79 | 8.39 | 10.98 | 11.53 |
| | R | 8.21 | 6.27 | 5.56 | 5.64 | 5.52 | 6.23 | 5.85 | 8.34 | 8.88 |
| DeepSeek-V3 | P | 9.46 | 7.69 | 6.82 | 8.34 | 7.69 | 7.97 | 8.72 | 10.39 | 10.95 |
| | R | 6.87 | 5.18 | 4.36 | 5.80 | 5.45 | 6.16 | 6.16 | 7.77 | 8.31 |
| Qwen-2.5 | P | 11.29 | 7.55 | 9.78 | 8.92 | 9.17 | 9.59 | 7.95 | 10.47 | 10.69 |
| | R | 8.64 | 5.05 | 7.18 | 6.35 | 6.59 | 7.00 | 5.43 | 7.85 | 8.06 |
| Mistral | P | 9.23 | 8.28 | 8.95 | 8.14 | 8.35 | 7.70 | 9.44 | 8.16 | 9.55 |
| | R | 6.65 | 5.74 | 6.38 | 5.61 | 5.81 | 5.19 | 6.85 | 5.63 | 6.96 |
| Gemma2 | P | 9.99 | 8.36 | 8.57 | 8.07 | 8.53 | 7.69 | 8.51 | 9.37 | 11.15 |
| | R | 7.38 | 5.82 | 6.02 | 5.54 | 5.98 | 5.18 | 5.96 | 6.78 | 8.51 |
| VITA-1.5 | P | 10.38 | 9.64 | 9.47 | 7.95 | 8.52 | 8.39 | 9.25 | 10.53 | 11.18 |
| | R | 7.76 | 7.04 | 6.88 | 5.43 | 5.97 | 5.85 | 6.67 | 7.90 | 8.53 |
| | | | | | Speech + Text | | | | | |
| ChatGPT-4o | P | 13.35 | 11.79 | 14.66 | 13.08 | 15.39 | 13.60 | 15.27 | 14.51 | 12.47 |
| | R | 10.98 | 9.45 | 12.27 | 10.72 | 12.99 | 11.23 | 12.87 | 12.12 | 10.12 |
| Gemini-1.5 | P | 13.63 | 13.12 | 13.93 | 13.35 | 14.15 | 13.87 | 14.88 | 12.96 | 13.01 |
| | R | 11.26 | 10.75 | 11.55 | 10.98 | 11.77 | 11.49 | 12.49 | 10.60 | 10.65 |
| Qwen2-Audio | P | 11.99 | 12.77 | 13.29 | 13.86 | 15.19 | 14.57 | 13.21 | 14.26 | 11.80 |
| | R | 9.64 | 10.41 | 10.92 | 11.48 | 12.79 | 12.18 | 10.84 | 11.88 | 9.46 |
| VITA-1.5 | P | 12.97 | 12.62 | 12.75 | 11.53 | 15.09 | 15.33 | 14.07 | 15.33 | 13.60 |
| | R | 10.61 | 10.26 | 10.39 | 9.19 | 12.70 | 12.93 | 11.69 | 12.93 | 11.23 |
| | | | | | Video + Text | | | | | |
| ChatGPT-4o | P | 18.82 | 16.99 | 19.71 | 19.28 | 18.47 | 20.18 | 18.71 | 21.19 | 20.24 |
| | R | 16.04 | 14.23 | 16.92 | 16.50 | 15.70 | 17.39 | 15.93 | 18.39 | 17.45 |
| Gemini-1.5 | P | 19.38 | 17.66 | 18.05 | 17.89 | 20.42 | 17.51 | 18.87 | 19.55 | 19.18 |
| | R | 16.60 | 14.89 | 15.28 | 15.12 | 17.63 | 14.75 | 16.09 | 16.76 | 16.40 |
| Qwen2.5-VL | P | 15.44 | 17.67 | 15.54 | 18.03 | 13.70 | 15.25 | 13.64 | 18.44 | 18.38 |
| | R | 18.58 | 16.03 | 17.12 | 14.52 | 13.95 | 17.25 | 17.51 | 15.12 | 16.88 |
| VITA-1.5 | P | 20.06 | 17.44 | 18.76 | 17.48 | 18.51 | 18.27 | 18.94 | 21.24 | 21.15 |
| | R | 17.27 | 14.68 | 15.98 | 14.72 | 15.73 | 15.50 | 16.16 | 18.44 | 18.35 |
| | | | | | Video + Speech + Text | | | | | |
| VITA-1.5 | P | 21.40 | 18.94 | 22.18 | 17.73 | 21.97 | 22.15 | 20.72 | 26.21 | 27.05 |
| | R | 18.95 | 16.51 | 19.72 | 15.31 | 19.52 | 19.67 | 18.27 | 23.73 | 24.57 |
| MM-When2Speak | P | **31.55** | **32.26** | **31.25** | **29.53** | **28.21** | **28.82** | **27.47** | **35.17** | **33.27** |
| | R | **29.24** | **29.95** | **28.94** | **27.22** | **25.91** | **26.52** | **25.17** | **32.85** | **30.95** |

sliding-window predictions exacerbate class imbalance (especially for silence), which in turn increases false prediction rates under our problem formulation.

We also report the latency of different representative models in Table 3. It shows that latency naturally increases with more modalities. Importantly, under each modality configuration, our model exhibits comparable or lower latency relative to baselines, while producing stronger accuracy. This demonstrates that MM-When2Speak remains practically feasible for real-time deployment.

Table 3: Latency comparison of different models across modalities in seconds.

| Modality | Text | | Text+Vision | | Text+Vision+Speech | |
|---|---|---|---|---|---|---|
| Models | Qwen-2.5 | Ours | Qwen2.5-VL | Ours | VITA-1.5 | Ours |
| Average | 0.103 | 0.092 | 0.561 | 0.606 | 1.132 | 1.145 |
| Min. | 0.068 | 0.073 | 0.217 | 0.235 | 0.779 | 0.698 |
| Max. | 0.388 | 0.369 | 0.835 | 0.825 | 1.889 | 1.732 |

Overall, results across both datasets highlight the consistent benefits of multimodal integration, with performance improving as additional modalities are introduced. By leveraging a sliding-window mechanism to handle long-form inputs, MM-When2Speak maintains robust performance in both short-clip and full-length scenarios. More comparisons with finetuned methods are provided in appendix Sec. D.

## 4.3 ABLATION STUDY

We report three representative ablation studies in the paper, while additional analysis, including model robustness to noisy inputs, cross-evaluation on the Behavior-SD Lee et al. (2025) dataset, comparisons with the backchannel detection method Amer et al. (2023), are provided in the appendix.

**Multimodality and Self-attention Fusion**. We conduct ablation studies to evaluate the effectiveness of multimodal inputs (Table 4). It can be seen that performance notably drops when only two

Table 4: Effectiveness of multimodal inputs of our MM-When2Speak on Short-Clips-Test.

| Metric | Affm. | Grat. | Fare. | Grtn. | Qstn. | Surp. | Pndr. | FR | Slnc. |
|---|---|---|---|---|---|---|---|---|---|
| (a) Ours (Video + Speech + Text) | | | | | | | | | |
| P | 62.21 | 64.35 | 63.15 | 63.29 | 46.26 | 50.52 | 37.78 | 68.15 | 68.78 |
| R | 59.86 | 61.99 | 60.79 | 60.44 | 43.91 | 46.25 | 35.45 | 65.79 | 66.42 |
| F1 | 61.01 | 63.15 | 61.95 | 62.09 | 45.05 | 49.62 | 36.58 | 66.95 | 67.58 |
| (b) Ours w/o Self-attention (Video + Speech + Text) | | | | | | | | | |
| P | 60.51 | 63.16 | 62.03 | 61.76 | 45.16 | 46.91 | 36.91 | 65.44 | 68.72 |
| R | 56.83 | 59.47 | 58.64 | 58.07 | 41.51 | 45.56 | 35.11 | 61.75 | 65.01 |
| F1 | 58.61 | 61.26 | 60.43 | 59.86 | 43.26 | 47.71 | 35.01 | 63.54 | 66.82 |
| (c) Ours (Video + Text) | | | | | | | | | |
| P | 36.89 | 38.65 | 41.09 | 38.83 | 34.22 | 32.71 | 31.75 | 43.49 | 42.32 |
| R | 44.22 | 44.41 | 40.45 | 42.93 | 36.07 | 33.72 | 24.12 | 46.05 | 41.49 |
| F1 | 40.23 | 41.64 | 40.85 | 40.94 | 33.66 | 33.22 | 27.72 | 44.71 | 41.89 |
| (d) Ours w/o Self-attention (Video + Text) | | | | | | | | | |
| P | 35.98 | 37.69 | 40.06 | 37.86 | 33.37 | 31.91 | 30.96 | 42.41 | 41.27 |
| R | 43.12 | 43.30 | 39.50 | 41.87 | 35.17 | 31.91 | 23.52 | 45.91 | 40.40 |
| F1 | 39.23 | 40.60 | 39.83 | 39.92 | 32.82 | 31.91 | 26.41 | 44.01 | 40.85 |
| (e) Ours (Speech + Text) | | | | | | | | | |
| P | 29.96 | 31.39 | 33.36 | 31.53 | 27.79 | 26.56 | 25.78 | 36.54 | 34.27 |
| R | 34.59 | 34.73 | 31.68 | 33.58 | 25.91 | 25.95 | 28.85 | 32.74 | 32.40 |
| F1 | 32.36 | 32.55 | 31.54 | 31.61 | 25.98 | 26.24 | 27.16 | 34.52 | 33.31 |
| (f) Ours w/o Self-attention (Speech + Text) | | | | | | | | | |
| P | 29.59 | 31.00 | 32.96 | 31.10 | 26.45 | 26.24 | 25.47 | 34.69 | 33.37 |
| R | 34.16 | 32.75 | 31.29 | 31.37 | 25.27 | 25.47 | 28.39 | 32.56 | 32.40 |
| F1 | 32.28 | 31.47 | 32.05 | 31.23 | 25.87 | 25.84 | 26.82 | 33.55 | 32.87 |

Table 5: Cross-domain performance comparison for MM-When2Speak and VITA-1.5. "In-domain" display model performance reported in Table 1.

| Method | Domain | Metric | Affm. | Grat. | Fare. | Grtn. | Qstn. | Surp. | Pndr. | FR | Slnc. |
|---|---|---|---|---|---|---|---|---|---|---|---|
| MM-When2Speak | In-domain | P | 62.21 | 64.35 | 63.15 | 63.29 | 46.26 | 50.52 | 37.78 | 68.15 | 68.78 |
| | | R | 59.86 | 61.99 | 60.79 | 60.44 | 43.91 | 46.25 | 35.45 | 65.79 | 66.42 |
| | Cross-domain | P | 54.23 | 58.64 | 56.95 | 60.55 | 41.51 | 45.16 | 31.05 | 64.29 | 60.24 |
| | | R | 51.69 | 53.82 | 58.39 | 57.75 | 39.74 | 42.98 | 30.68 | 63.34 | 61.17 |
| VITA-1.5 | In-domain | P | 38.03 | 39.68 | 39.16 | 38.81 | 28.46 | 31.23 | 23.32 | 31.36 | 35.32 |
| | | R | 35.14 | 36.79 | 36.27 | 35.92 | 25.61 | 28.37 | 20.50 | 28.50 | 32.44 |
| | Cross-domain | P | 24.55 | 30.02 | 27.45 | 29.47 | 19.96 | 26.44 | 21.09 | 35.70 | 31.12 |
| | | R | 20.68 | 22.36 | 25.86 | 27.04 | 18.78 | 22.54 | 16.69 | 22.38 | 27.37 |

modalities are used, and replacing video with speech further degrades results. This highlights the complementary role of all three modalities in achieving robust response-type prediction. Additionally, all models display performance degradation as we remove the self-attention, indicating the importance of our self-attention mechanism.

To further analyze class-wise effects, we present confusion matrices for the ablation settings (Fig. 3). Adding more modalities consistently improves diagonal dominance, with the best performance observed when all are combined. Notably, video modality yields denser diagonal distributions, and using self-attention mitigates mispredictions, underscoring the importance of visual signals and proper multimodal feature fusion for capturing subtle conversational cues.

**Cross-domain Evaluation**. We evaluate the generalizability of MM-When2Speak through cross-dataset experiments on our Short-Clips dataset, which consists of two distinct subsets (News and Zoom meetings) representing different conversational scenarios. The model is trained on the News subset and tested on the Zoom subset without overlap, and results are reported in Table 5 using precision and recall. For comparison, we also present in-domain results as well as the VITA-1.5 baseline, which is evaluated under the same cross-domain setting. Although cross-dataset evaluation shows a performance drop compared to in-domain results, it remains smaller than that of VITA-1.5, which is reasonable given the domain shift. Crucially, MM-When2Speak still outperforms VITA-1.5 across all metrics under cross-domain settings, demonstrating its strong generalizability.

**Size and Stride of Sliding Window**. We use a sliding window mechanism on our model for full video inferences, where we use a window size of 10 seconds and a stride of 0.5 seconds. We ablate these two parameters to measure their impact on performance on Full-Videos dataset. We follow the same training/testing strategy introduced earlier, and the results are reported in Table 6 and 7. Notably, performance peaks when opting window size is 10s and stride is 0.5s, which verifies our

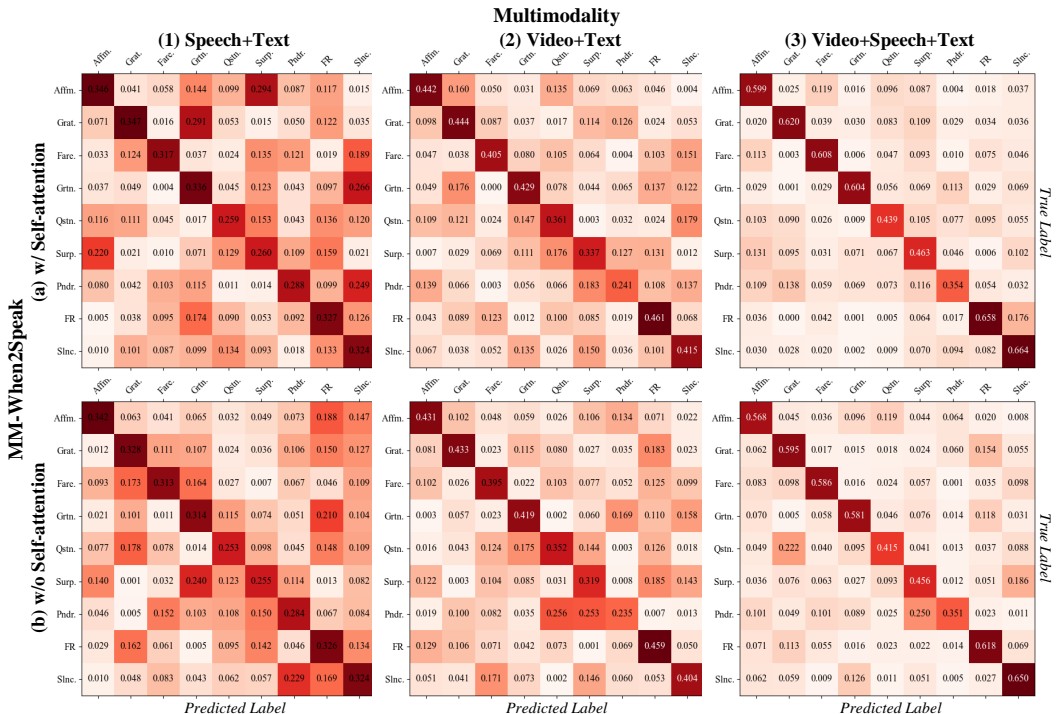

Figure 3: Confusion matrices of MM-When2Speak under different modality configurations, with and without the proposed self-attention fusion mechanism.

==design. This also suggests that using sliding window with appropriate size and stride may include sufficient multimodal contexts for accurate response type predictions.==

==Table 6: Performance of MM-When2Speak with different window sizes (sliding stride=0.5s).==

| Window Size | Metric | Affm. | Grat. | Fare. | Grtn. | Qstn. | Surp. | Pndr. | FR | Slnc. |
|---|---|---|---|---|---|---|---|---|---|---|
| 1s | P | 27.42 | 28.36 | 27.15 | 34.41 | 24.55 | 25.71 | 24.02 | 31.02 | 30.12 |
| | R | 25.11 | 26.12 | 25.02 | 22.14 | 22.14 | 23.11 | 22.31 | 28.14 | 27.11 |
| 2s | P | 28.14 | 29.57 | 28.33 | 36.22 | 26.84 | 25.11 | 25.11 | 32.14 | 31.44 |
| | R | 26.42 | 27.14 | 26.11 | 24.88 | 23.57 | 24.02 | 23.88 | 29.01 | 28.63 |
| 5s | P | 30.44 | 31.44 | 30.66 | 38.11 | 27.41 | 27.91 | 26.71 | 33.77 | 32.55 |
| | R | 28.71 | 28.94 | 27.55 | 25.94 | 24.88 | 25.77 | 24.72 | 30.72 | 29.66 |
| **10s** | P | **31.55** | **32.25** | **31.25** | **39.53** | **28.21** | **28.82** | **27.47** | **35.17** | **33.27** |
| | R | **29.24** | **29.95** | **28.94** | **27.22** | **26.52** | **25.17** | **25.17** | **32.85** | **30.95** |
| 15s | P | 29.41 | 30.52 | 29.14 | 37.08 | 26.55 | 27.14 | 26.04 | 33.66 | 32.22 |
| | R | 27.55 | 28.44 | 27.01 | 24.66 | 24.12 | 25.01 | 24.22 | 31.44 | 29.40 |
| 20s | P | 27.95 | 28.78 | 27.66 | 35.22 | 25.41 | 25.94 | 24.91 | 31.84 | 30.47 |
| | R | 26.17 | 26.91 | 25.55 | 23.55 | 23.01 | 23.72 | 23.11 | 29.77 | 28.52 |

==Table 7: Performance of MM-When2Speak with different window sliding strides (window size=10s).==

| Sliding Stride | Metric | Affm. | Grat. | Fare. | Grtn. | Qstn. | Surp. | Pndr. | FR | Slnc. |
|---|---|---|---|---|---|---|---|---|---|---|
| 0.1s | P | 28.15 | 29.24 | 28.18 | 34.96 | 24.74 | 25.34 | 24.62 | 31.13 | 30.12 |
| | R | 25.83 | 26.36 | 25.86 | 24.51 | 23.23 | 23.29 | 23.11 | 29.18 | 28.48 |
| 0.2s | P | 29.76 | 30.98 | 29.88 | 36.66 | 26.25 | 26.64 | 26.02 | 33.13 | 30.66 |
| | R | 27.34 | 27.52 | 26.91 | 25.05 | 24.80 | 25.04 | 23.48 | 30.27 | 28.68 |
| **0.5s** | P | **31.55** | **32.25** | **31.25** | **39.53** | **28.21** | **28.82** | **27.47** | **35.17** | **33.27** |
| | R | **29.24** | **29.95** | **28.94** | **27.22** | **25.91** | **26.52** | **25.17** | **32.85** | **30.95** |
| 1s | P | 29.59 | 29.73 | 29.00 | 37.18 | 25.88 | 25.88 | 25.87 | 33.38 | 30.83 |
| | R | 27.09 | 27.97 | 26.63 | 25.59 | 24.20 | 25.19 | 23.11 | 30.02 | 29.14 |
| 2s | P | 27.28 | 27.73 | 27.87 | 35.31 | 25.10 | 25.47 | 24.65 | 30.91 | 28.66 |
| | R | 25.72 | 26.59 | 25.57 | 23.69 | 22.93 | 23.28 | 22.77 | 29.14 | 27.52 |
| 5s | P | 25.45 | 26.25 | 24.84 | 30.96 | 22.10 | 22.44 | 21.46 | 27.98 | 26.03 |
| | R | 24.08 | 23.66 | 23.39 | 21.72 | 20.19 | 21.59 | 19.78 | 26.80 | 24.69 |

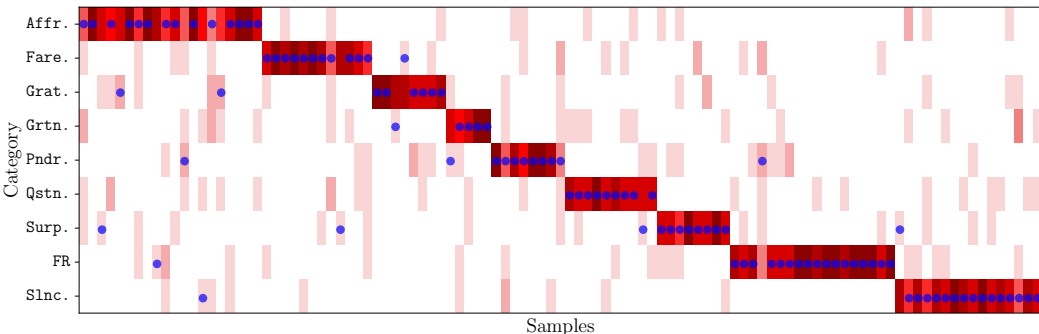

Figure 4: Annotation comparison between our pipeline-generated and human manual labels. The red heatmaps indicate the votes of how participated volunteers label each sample, while the blue circles denote the ChatGPT generated labels. 87.62% of the sample labels are matched up, validating our dataset quality.

## 4.4 HUMAN STUDY ON SHORT-CLIPS DATASET

As in Sec.3.1, we use ChatGPT to determine the specific reaction type for each reaction segment in our dataset. To assess how much predictions are aligned with human classifications so as to verify annotation quality, we conduct a small-scale human study on Short-Clips data classifications for verification. We sample a total of 105 random samples of our Short-Clips dataset, and ask nine volunteers to label each 10s video (with audio and text) using one of the nine response types. Note that the study is designed to evaluate the reliability of our automatic annotation pipeline. 105 clips are formed with stratified sampling over classes and domains to cover both frequent and long-tail categories. Testers receive the same written class definitions used by the pipeline but no examples from the test set. Each tester independently label all 105 clips with one of the nine categories. Annotators are blinded to both the pipeline outputs and to other annotators' decisions, with no discussion permitted. We do not enforce consensus or adjudication, as the goal is to measure alignment between the pipeline and a diverse set of human judgments.

After obtaining all manually-labeled results, we compare them with our pipeline-generated annotations, and the results are shown in Fig. 4. We assess the reliability of model predictions by aligning them with aggregated human annotations. Specifically, the red heatmap represents the distribution of human votes for each category, where darker regions indicate stronger consensus. The blue circles denote the generated labels for each sample. A prediction is regarded as reliable if the corresponding blue circle falls within the darkest region along the vertical axis for that sample, i.e., the category receiving the highest proportion of human votes. It can be seen that 92 out of 105 (87.62%) samples are correctly categorized by ChatGPT compared to human labels, suggesting that the annotation pipeline we adopted is able to produce relatively reliable labels as humans, which also proves the quality of our dataset.

More information and results (e.g., information of the voluntary participants, inter-annotator agreements) of this human study can be found in the appendix Sec. H.

## 5 CONCLUSION

In this work, we propose MM-When2Speak, a multimodal LLM designed to predict when an AI conversational agent should respond in a human conversation. By reformulating this task as a dense, window-based classification problem, MM-When2Speak effectively integrates multimodal signals (video, speech, text), to capture fine-grained conversational cues essential for determining "when to speak", particularly "when to react". We construct two curated datasets that span conversational interactions and demonstrate that our method outperforms a wide range of LLMs across various evaluation settings. Extensive ablation studies further highlight the advantages of multimodal learning and our self-attention fusion strategy for conversational cues. In the future, we aim to extend MM-When2Speak to larger and more diverse datasets, multi-party group conversations, and other conversational settings such as presentations and meetings, where modeling more complex conversational dynamics will be essential for real-world deployment.

## REPRODUCIBILITY STATEMENT

Along with a human study in the main paper, we have provided a detailed description of the dataset collection, preprocessing, and annotation procedures in the supplementary material to ensure transparency. To further facilitate reproducibility and follow-up research, we will release the dataset and source code publicly upon acceptance of the paper. This will enable other researchers to conduct further studies and build upon our work.

## LLM USAGE STATEMENT

We employed LLM to assist in some parts of the writing process, specifically for final linguistic refinement and content compression. The model was only used to improve clarity and conciseness of the text, helping us meet page limits, without influencing any of the technical content, methodology, or results of the paper.

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

# APPENDIX

The following are the appendix contents for our main paper.

- In **Sec. A**, we describe the process of collecting and pre-processing of the videos from online sources, then specify the label definitions for our curated datasets using audio diarization, especially the detailed explanation for our Full Video dataset.
- In **Sec. B**, we present the architectural details of our MM-When2Speak, and two representative qualitative examples of how multimodalities contribute different to accurate response type predictions.
- In **Sec. C**, we provide the prompt template we use to conduct response type predictions, and display some sample outputs along with success/failure predictions case studies.
- In **Sec. D**, we further compare our method with several finetuned methods across different modalities, including Qwen2.5 (text), Qwen2.5-VL (text+vision), Video-ChatGPT (text+vision) and VITA-1.5 (text+vision+speech).
- In **Sec. E** we verify the robustness of our method when dealing with degraded inputs, such as noisy audio and downgraded images.
- In **Sec. F** we conduct an extra cross-evaluation ablation on a speech+text synthesized text+speech dataset Behavior-SD Lee et al. (2025) to validate the generalizability of our method.
- In **Sec. G**, we conduct an comparison experiment with backchannel detection method One Stream.
- In **Sec. H**, we provide details (i.e., information of voluntary participants, inter-annotater agreement statistics) of our human study on our dataset annotations.
- In **Sec. I**, we compare the performance difference of using Qwen2-7B-base and Qwen2-7B-instruct as the LLM in our model to evaluate the impact of instruction tuning.
- In **Sec. J**, we discuss issues including multi-party conversations, model extension to "how to speak", and comparison with temporal/event-boundary methods, providing insights of our design.

## A  DATASET SPECIFICATIONS

### A.1  VIDEO COLLECTION AND PREPROCESSING

It is noteworthy that **our research mainly focuses on dyadic conversation scenarios** like video digital chatbot, and we limit them to where **two speakers can clearly see each other's face and hear each other's voice**, while having cameras capturing their **frontal faces** at relatively the same height level. This two-person frontal-face conversation setup ensures a **clear and controlled setting for learning core interaction dynamics**, such as frontal facial expressions and head movements that reflect conversational signals like reactions, turn-takings, responsiveness, and behavioral coordinations, which could be used to extend to more complex conversational scenarios such as multi-group conversations. With that, we collected video data of two basic categories: **news discussions (e.g., CNN, MSNBC, etc.), and Zoom meetings**. In these videos, two conversational participants are separated into split-screen frames, with one shown on the left and the other on the right (or up and down, respectively), engaging in turn-taking discussion of the topics while facing to the cameras. We construct a script to download this kind of videos using yt dlp (2023), and gather them together forming a raw dataset with 2,403 videos collected.

The objective of data collection is to collect conversational clips that contain **clean, consistent, and high-quality** visual and audio data. However, not all downloaded videos are usable since a lot of them suffer from some extreme low-quality factors, such as long-time black screen, regular scene change, noise interferences, or severe sound overlaps, which hinder the quality of the data of dyadic conversations. Therefore, we conduct further analysis of all downloaded videos to filter out those with such unbearable situations. We manually review each video to examine if any of the above factors exist in each video, and eventually come up with **377 high-quality videos**.

It is worth noting that the 377 collected videos exhibit significant variation in duration, ranging from **1 minute 54 seconds** to **9 minutes 43 seconds**. Moreover, **not all segments within each video are suitable for dyadic interaction analysis**. For instance, some videos begin with only a single speaker, and then involve the second speaker. To ensure the dataset focuses exclusively on actual dyadic

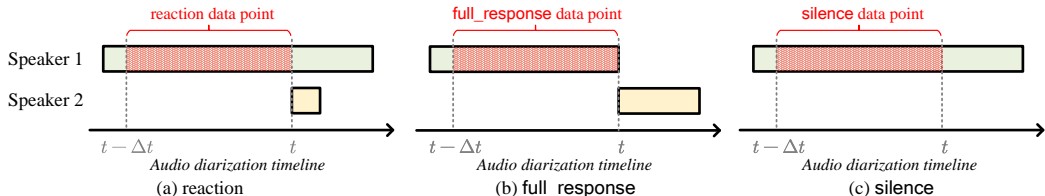

Figure 5: Illustration of short clip data point obtainment. We determine the data points by analyzing the audio diarization results. The green and yellow bars denote the diarization results of two different speakers' speaking time in one collected video. (a), (b), and (c) represent the typical data point sampling scenarios for reaction, full_response, and silence. Note that in this illustration, when speaker 1 is speaking and does not intend to stop, speaker 2 is the listener. The silence data point can be sampled anywhere whenever there is only one speaker speaking. $\Delta t$ denotes window size.

conversations, we manually crop each video to extract only the relevant segments. The resulting 377 cropped videos serve as the foundation of our Short Clips dataset, whose **average time is 2 minutes 13 seconds**.

## A.2 DATA ANNOTATION

For the video modality data, we crop out the **rectangular face area** of each speaker in each video first. For the speech and text modalities, we employ **audio diarization** to facilitate the annotation. That is, we extract the audio of each high-quality video, and diarize the audio into two split audios by speakers using Asteroid (2023); pyannote (2020), where each split audio represents only one speaker speaking, along with the timestamps indicating whenever they start and stop speaking. We then adopt Whisper OpenAI (2023) to **transcribe** the diarized audio into texts with timestamps grouped by speakers. Based on the timestamps, we manage to define the three basic categories (i.e., reaction, full_response, and silence), where:

1. **reaction**: when a listener speaks for a very short time and does not stop the speaker speaking, we consider that a "reaction" data point, as shown in Fig. 5(a);

2. **full_response**: when a listener starts speaking and the speaker stops for them to speak, we consider that a "full_response" data point, as shown in Fig. 5(b);

3. **silence**: when the speaker continues speaking and the listener has no intention of speaking, we consider that a "silence" data point, as shown in Fig. 5(c);

Based on the category information, we extract $\Delta t$ seconds (10 seconds in default) multimodal (i.e., video, speech, and text) short clips prior to the corresponding timestamp, as one data point in the Short Clips dataset. Note that we can obtain timestamps on a per-word basis, so we are able to finely extract texts with that window.

Furthermore, we proceed to classify reactions into **seven fine-grained categories**. We first gather all texts from those short clips labeled with "reaction", and feed them into ChatGPT OpenAI (2023) to classify into different reaction categories, which finally results in seven different categories: affirmation, gratitude, greeting, farewell, surprise, question, and pondering. We then assign these fine-grained reaction labels back to those short clips, and eventually come up with nine categories for response-type prediction.

For the generation of the seven reaction categories, we first organize all texts of reaction data into a text file, with each line representing one reaction data point. The following is a sample of the text file:

```
Yeah, that's right.
Oh, really?
Hmm.
Wow!
Thank you.
Absolutely.
...
```

And then, we upload this text file to ChatGPT, and use the following prompt to obtain the seven reaction categories as well as the fine-grained category for each line in the text file. The results are saved to a newly created text file, which contains both the summarized categories and the specific reaction category assigned to each line of the uploaded text file. We also **manually examine** the resulting categorization of the reaction data to ensure no explicit outlier (e.g., misclassification from `affirmation` to `question`).

```
You are a linguistic analysis assistant. Your task is to analyze a list of short utterances, where
each line in the uploaded text file represents a brief reaction from a speaker during a conversation.
These reactions belong to different "reaction types" (e.g., Affirmation, Surprise, etc..), which
you need to summarize and categorize solely based on the uploaded text file.

Please perform the following tasks:

1. Read all the utterances in the text file;
2. Identify and list all distinct **reaction types** found in the text, provide a brief explanation
for each type, and make sure these types make sense;
3. Assign a corresponding reaction type label to each line of text;
4. Output your results to a new text file in the following format:
- A list of identified reaction types with explanations
- The original line followed by its assigned label

For example, if the input file contains the following four lines (within the double quotation marks):

"Yes, I agree with you.
Oh, really?
Hmm.
Are you sure? "

You should output the following in a new text file:

Identified Reaction Types:
- Affirmation: indicates agreement or confirmation
- Surprise: expresses shock, amazement, or unexpectedness
- Pondering: the process of thought according to what the speaker says
- Question: expresses doubts or raising concerns

Classification Results:
- Yes, I agree with you. -> Affirmation
- Oh, really? -> Surprise
- Hmm. -> Pondering
- Are you sure? -> Question

Now, please perform the same analysis based on the uploaded text file.
```

Based on the output from ChatGPT, we summarize seven different reaction types, which are `affirmation`, `gratitude`, `greeting`, `farewell`, `surprise`, `question`, and `pondering` respectively. Note that we have included a human study assessing the quality of such method to generate annotations for reaction data point in our main paper (Sec. 4.4) as well as in the appendix (Sec. H). The results demonstrate that differences between human efforts and this annotation pipeline is modest rather than dramatic, suggesting that this labeling approach is comparatively reliable.

## A.3 DATASET COMPOSITION

For a more comprehensive evaluation of our model, we compose our dataset with **two different subsets**: One subset, containing sampled 357 out of the entire collected videos, is segmented into a set of $\Delta t = 10s$ **short clips**, serving to validate the model's ability to perform accurate response type prediction on short, fixed-length multimodal segments; The other subset was kept as **full videos**, which contain long-range full conversations between two speakers from the remaining 20 videos, where we employed a sliding-window strategy to simulate real-world application settings and to assess the model's performance over complete conversational scenarios.

### A.3.1 SHORT-CLIPS DATASET

For the 357 collected videos used for Short-Clips dataset, based on identified reaction types, we obtain 2,433 data points for `affirmation`, 820 for `gratitude`, 253 for `farewell`, 244 for `greeting`, 192 for `question`, 289 for `surprise`, and 162 for `pondering`, which construct 4,393 reaction data points altogether. We also randomly sample 2,000 `full_response` and 2,000 `silence` data points.

Then, we use a train/test ratio of 7:3 to split each response type data point (including specific reaction types) using stratified sampling, and construct the Short-Clips-Train and Short-Clips-Test datasets.

### A.3.2 FULL VIDEOS DATASET

Based on the same pipeline as described above, we use the remaining 20 videos independent of our Short Clips dataset to evaluate methods of their ability to classify response types, which resembles real-world long-term response type predictions based on a sliding window mechanism. These videos are annotated the same way as in Sec. A.2. The average time for these videos is **2 minutes 25 seconds**. In these 20 videos, we have 52 reaction data points (including 52 `affirmation`, 16 `gratitude`, 8 `farewell`, 10 `greeting`, 28 `surprise`, 37 `question`, and 16 `pondering`), and 98 `full_response` data points.

We use the sliding window to slide across the video with a window size of $\Delta t = 10s$ and a stride of $\delta = 0.5s$, and extract the timestamps of the segments captured by the sliding window. Unlike annotating Short-Clips dataset, the use of strided sliding window can introduce slight time misalignments with the diarization timestamps. To mitigate this task-related uncertainty and ambiguity, we examine the timestamps of the window with the annotations corresponding to the video:

- If the **end time** (i.e., the **right end of the sliding window**) does not overlap with any of the response-type annotation, the windowed clip will be assigned a "silence" label;
- otherwise, it will be determined by the **overlapped data point**, such as a specific reaction type or full response.

To more effectively assign the data point label to the correct windowed clip, we use a threshold of $\tau = 250$ milliseconds to define a **temporal neighborhood** of a given response-type data point at time $t$. That is, when the end time of a windowed clip falls in the neighborhood $[t - \tau, t + \tau)$, we assign the label of the data point at time $t$ to the windowed clip, which serves as **a tolerance for labeling as well as prediction**. Note that for transcripts, audio diarization provides word-level timestamps. A word is included in a window if its starting timestamp falls within the window's $[t - \Delta t, t)$ ($\Delta t$ is the window size) interval, **regardless of overlapping**. This ensures strict temporal consistency across video, audio, and text streams.

Using this sliding window mechanism, we obtain 5,482 clips from these 20 videos, constructing a dense response-type prediction task to resemble "when to speak" problem in real-world conversations.

## B ARCHITECTURAL DETAILS FOR MM-WHEN2SPEAK

To better model the relative importance of information across modalities, we apply a lightweight self-attention mechanism directly on the concatenated multimodal embeddings before classification. Specifically, given a sequence of visual, audio, and textual embeddings extracted from the respective encoders, we first project all token embeddings into a shared latent space of 4096 dimensions. In our experiments, the visual modality consists of 10 frames, each producing 256 tokens via the visual encoder, resulting in a total of 2560 visual tokens. The audio encoder outputs 361 audio tokens for a 10-second segment, and the text encoder provides $X$ textual tokens, depending on the utterance tokenization length. These three modalities are concatenated along the sequence dimension to form a unified sequence of $(2921 + X)$ tokens, each with 4096-dimensional embeddings.

We then apply a single-layer token-wise self-attention module over this concatenated sequence to enable early-stage cross-modal interaction. This mechanism allows each token to attend to all other tokens, regardless of their modality, and dynamically reweights them based on relevance to the response-type prediction task. Unlike modality-specific fusion strategies or hard-coded weighting, this design leverages the inherent flexibility of attention to discover salient features across modalities and time. Importantly, because attention weights are computed at the token level solely based on inputs, we can infer which parts of the input contribute most to the model's final prediction.

Fig. 7 and Fig. 8 illustrate how the self-attention mechanism allocates weights across the concatenated sequence of multimodal inputs. In Fig. 7, the text alone does not clarify if the speaker has finished. However, the last video frame clearly shows the speaker's mouth is still open, indicating ongoing speech and that the correct response type should be "silence". As the figure illustrates, our model

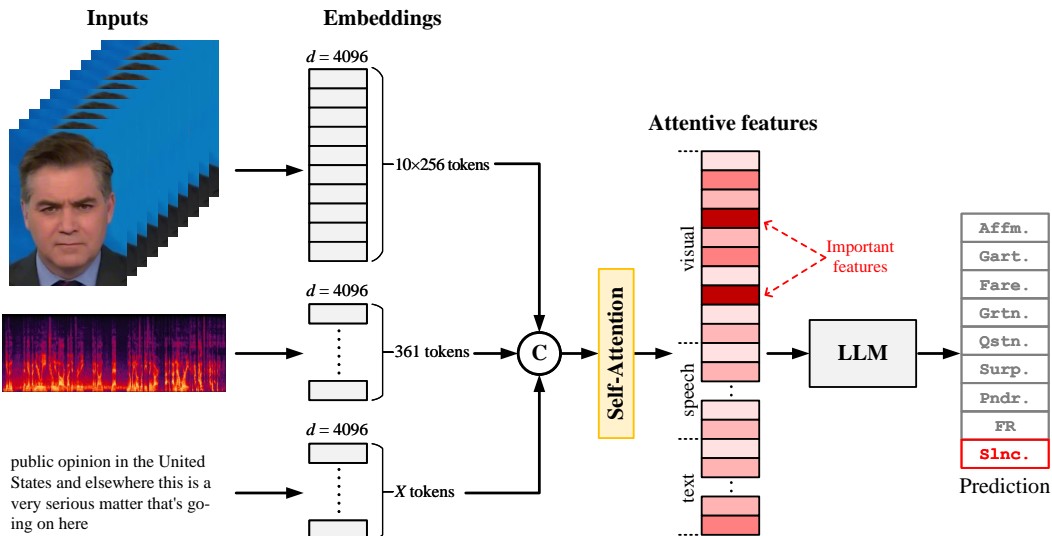

Figure 6: Feature fusion strategy for our proposed MM-When2Speak. The self-attention mechanism is used to enhance critical features in the concatenated embedding that facilitate accurate response-type predictions.

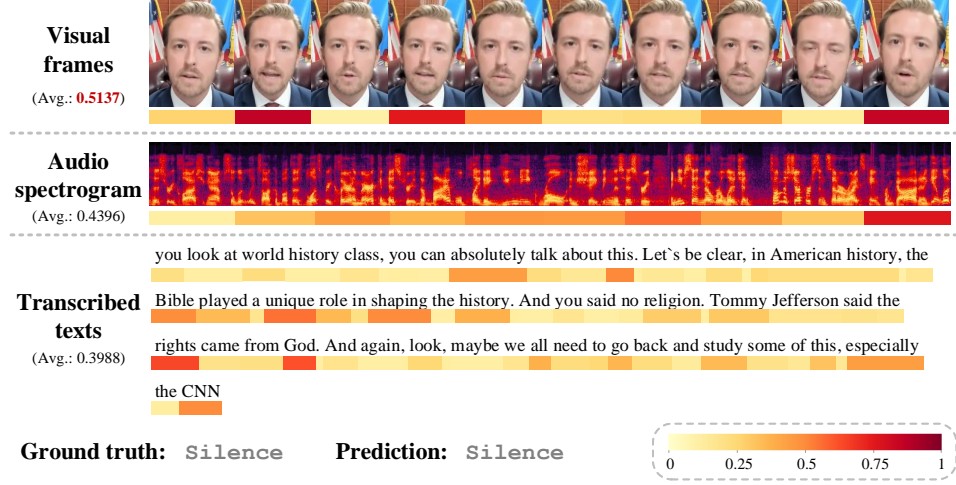

Figure 7: Visualization of token weights in the self-attention. Note that we average token weights by frames for video and speech modalities. We also report the modality-wise average attention weights, and it can be seen that the visual modality contains relatively more important information with an average attention weight of 0.5137.

placed significant attention on this final video frame with an average attention weight of 0.5137 for the visual modality, resulting in the correct "silence" prediction. On the other hand, Fig. 8 presents an example where the visual modality is ambiguous regarding speaker's completion. Here, the audio and text modalities clearly indicate that the speaker has finished. Consequently, our model primarily attends to the audio and text modalities, leading to the correct "full_Response" prediction.

## C PROMPT TEMPLATE FOR MODEL EVALUATION AND CASE STUDY

We use the following prompt to instruct a large multimodal language model (e.g., ChatGPT-4o , Gemini-1.5, Qwen2.5, etc.) to perform a 9-way classification task on short conversational clips. Each input sample consists of a 10-second multimodal clip, including a frontal facial video of the speaker,

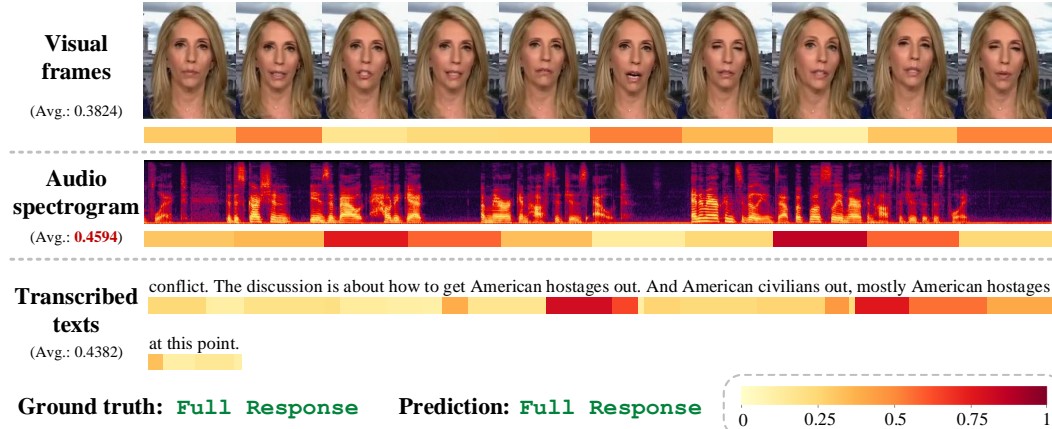

**Visual frames**
(Avg.: 0.3824)

**Audio spectrogram**
(Avg.: **0.4594**)

**Transcribed texts**
(Avg.: 0.4382)

conflict. The discussion is about how to get American hostages out. And American civilians out, mostly American hostages at this point.

**Ground truth: Full Response**    **Prediction: Full Response**

0    0.25    0.5    0.75    1

Figure 8: Visualization of token weights in the self-attention. Note that we average token weights by frames for video and speech modalities. We also report the modality-wise average attention weights, and it can be seen that the speech modality contains relatively more important information with an average attention weight of 0.4594.

their corresponding audio signal, and the exact transcript of spoken words. The model is asked to holistically assess the speaker's verbal content, prosody, and nonverbal facial cues to determine what type of response, if any, should follow immediately after the segment. The possible target categories include brief reactions (i.e., `affirmation`, `gratitude`, `farewell`, `greeting`, `surprise`, `question`, and `pondering`), initiating full assistant replies (i.e., `full_response`), or maintaining silence (i.e., `silence`). The prompt enforces **strict output formats** by requiring the model to respond with exactly one of nine predefined lowercase labels and to avoid any explanatory text. This setup ensures consistency and interpretability of the model's predicted reaction types across diverse conversational contexts.

```
You are an intelligent multimodal assistant (breifly "assistant" below). Your task is to classify
each 10-second segment of user input into exactly one of nine predefined conversational reaction
types. Each input consists of three synchronized modalities:

- A video recording of a human speaker facing directly to the camera;
- An audio clip containing the speaker's voice during the same 10 seconds;
- A transcript of the exact words spoken by the user during that time window.

Your goal is to analyze the entire 10-second input holistically, and determine what type of response,
if any, should follow immediately after this segment ends. You must choose one and only one of the
following nine response categories:

1. affirmation: a short verbal confirmation or agreement (e.g., "yeah", "I see");
2. gratitude: a brief expression of thanks (e.g., "thanks", "appreciate it");
3. farewell: a polite ending or goodbye (e.g., "bye", "see you");
4. greeting: a short greeting or acknowledgment (e.g., "hi", "hello");
5. surprise: an exclamation of unexpectedness or amazement (e.g., "wow", "oh my god");
6. question: a short clarifying or curious reaction in the form of a question (e.g., "are you
sure?");
7. pondering: a thoughtful pause or reflective sound (e.g., "hmm", "well");
8. full_response: indicates that the user has likely completed speaking, and the assistant should
begin a full verbal response;
9. silence: no response should be given; the assistant should remain silent and let the user continue
speaking;

The following are some important constraints you need to pay full attention to:

- This is strictly a 9-class classification task.
- You must rely on all three modalities: video of the user's face, speech audio, and transcribed
text.
- You must select exactly one label per input.
- Your output must contain only the label name, written in all lowercase letters, and must match
exactly one of the categories listed above.
- Do not provide any explanation, reasoning, justification, or extra text.
```

```
    Now, given the following multimodal input, classify it into one of the nine categories above. Output
    only the category name.
```

Generally, the LLM will output a short message of "ready for input" to indicate that it is ready for the input afterwards, such as:

```
    Understood. I will classify the input multimodal clip you provide for me from now. Now please upload
    the 10-second multimodal input for classification.
```

The following are three examples of prediction of reaction (`affirmation`, as shown in Fig. 9 and Table 8), `full_response` (as shown in Fig. 10 and Table 9), and `silence` (as shown in Fig. 11 and Table 10), based on the prompt template described above, which is used to initialize the inference process of LLMs. We present the output of ChatGPT-4o, Qwen2.5 for text-only modality; Gemini-1.5 for speech+text modalities; Qwen2.5-VL for video+text modalities; VITA-1.5 for video+speech+text modalities. We also include our MM-When2Speak for comparisons. Compared to other methods, the results demonstrate that our proposed MM-When2Speak predicts the response-type labels accurately and consistently.

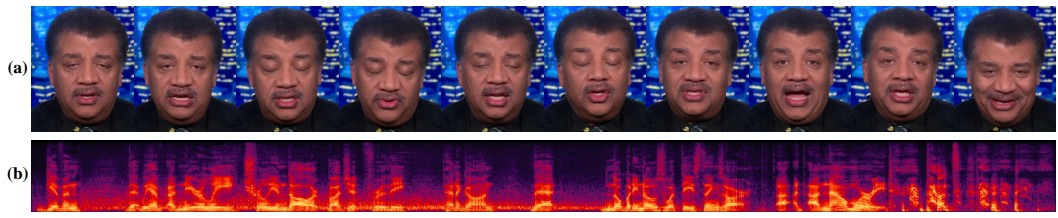

(a)

(b)

(c) But I'm a little disappointed that, given that we have a nearly trillion dollar budget for the Pentagon, that nobody knows what these things are. Now I'm worried about

Figure 9: A sample short clip of reaction label "`affirmation`". (a), (b), and (c) denote video frames, speech spectrogram, and transcribed texts, respectively.

Table 8: Prediction results of the Fig. 9 multimodal input from different LLMs. The ground truth is a reaction type "`affirmation`".

| Model | Text-only | | Speech + Text | Video + Text | Video + Speech + Text | |
|---|---|---|---|---|---|---|
| | ChatGPT-4o | Qwen2.5 | Gemini-1.5 | Qwen2.5-VL | VITA-1.5 | MM-When2Speak |
| Cls. | surprise | question | surprise | surprise | affirmation | affirmation |

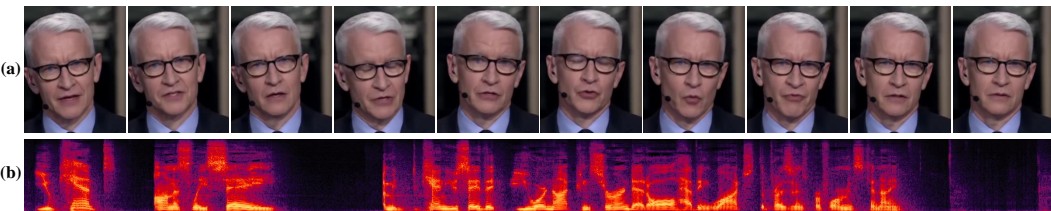

(a)

(b)

(c) you can't honestly say, I mean, can you say that you are not concerned at all having watched the president's performance tonight?

Figure 10: A sample short clip of label "`full_response`". (a), (b), and (c) denote video frames, speech spectrogram, and transcribed texts, respectively.

Table 9: Prediction results of the Fig. 10 multimodal input from different LLMs. The ground truth is "`full_response`".

| Model | Text-only | | Speech + Text | Video + Text | Video + Speech + Text | |
|---|---|---|---|---|---|---|
| | ChatGPT-4o | Qwen2.5 | Gemini-1.5 | Qwen2.5-VL | VITA-1.5 | MM-When2Speak |
| Cls. | full_response | silence | silence | silence | full_response | full_response |

(a)

(b)

(c) I've been paying close attention to legal experts who have been commenting on this. Um, you know, I always go back to first principles, and, and, look, I'm not a support of Donald Trump. I've been very clear about that.

Figure 11: A sample short clip of label "silence". (a), (b), and (c) denote video frames, speech spectrogram, and transcribed texts, respectively.

Table 10: Prediction results of the Fig. 11 multimodal input from different LLMs. The ground truth is "silence".

| Model | Text-only | | Speech + Text | Video + Text | Video + Speech + Text | |
|---|---|---|---|---|---|---|
| | ChatGPT-4o | Qwen2.5 | Gemini-1.5 | Qwen2.5-VL | VITA-1.5 | MM-When2Speak |
| Cls. | affirmation | affirmation | affirmation | silence | silence | silence |

Notably, natural conversational reactions are **inherently imbalanced**: high-frequency categories (e.g., Silence, Full Response) dominate real interactions, while nuanced reactions (e.g., Pondering) and boundary events (e.g., Farewell) form the natural long tail. In our Short-Clips dataset, Affirmation and Gratitude account for 55.39% and 18.66%, whereas Question and Pondering are $< 5\%$, and Surprise/Greeting/Farewell occur in the 5–7% range. Such skew is expected in open-domain human dialogue and correlates with lower human–human agreement. Regarding case characteristics, our qualitative analysis shows:

**Successful predictions tend to occur when:**

- Modalities reinforce each other (prosody, lexical cues, facial expressions align), e.g., positive tone + smile → Affirmation.

- Visual cues are salient, e.g., widened eyes or abrupt mouth movement → Surprise/Question.

- Clear transition cues exist in speech, such as explicit queries or turn boundaries → Full Response.

**Failure cases generally arise when:**

- Signals are weak or ambiguous, such as minimal facial change or monotone speech, often causing the model to default to Silence.

- Modalities contradict each other, e.g., smiling during sarcasm, causing confusion between Affirmation and Surprise/Question.

## D    COMPARISON WITH FINETUNED METHODS

In this section, we compare the performance of our proposed MM-When2Speak with different finetuned methods across modalities to verify its effectiveness. We train each model on our Short-Clips-Train dataset and test on Short-Clips-Test dataset following the same setting in our paper.

### D.1    SINGLE MODALITY: QWEN2.5 (TEXT)

We report the performance of pretrained and fintuned Qwen2.5 Yang et al. (2024b) compared to our proposed MM-When2Speak in Table 11. It can be seen that finetuning improves overall performance, but there exists significant gaps to our model, which suggests only using text modality for addressing "when to speak" may not be sufficient.

Table 11: Performance evaluations of response type prediction on Short-Clips-Test dataset for Qwen2.5 and MM-When2Speak. P: precision, R: recall. †: Model fintuned on Short-Clips-Train. Our MM-When2Speak is triple modalities.

| Model | Metric | Affm. | Grat. | Fare. | Grtn. | Qstn. | Surp. | Pndr. | FR | Slnc. |
|---|---|---|---|---|---|---|---|---|---|---|
| | | | | | Text | | | | | |
| Qwen-2.5 | P | 19.74 | 16.76 | 14.37 | 13.05 | 13.03 | 10.76 | 14.39 | 20.64 | 21.88 |
| | R | 16.44 | 13.51 | 11.17 | 9.89 | 9.87 | 7.68 | 11.19 | 17.33 | 18.55 |
| Qwen-2.5† | P | 22.50 | 18.69 | 14.98 | 15.93 | 13.13 | 12.05 | 14.79 | 21.83 | 24.10 |
| | R | 19.38 | 15.33 | 12.50 | 10.15 | 12.91 | 8.84 | 12.05 | 19.82 | 20.45 |
| MM-When2Speak | P | **62.21** | **64.35** | **63.15** | **63.29** | **46.26** | **50.52** | **37.78** | **68.15** | **68.78** |
| | R | **59.86** | **61.99** | **60.79** | **60.44** | **43.91** | **46.25** | **35.45** | **65.79** | **66.42** |

## D.2 DOUBLE MODALITIES: QWEN2.5-VL AND VIDEO-CHATGPT (VISION+TEXT)

We report the performance of Qwen2.5-VL Bai et al. (2025) and Video-ChatGPT Maaz et al. (2024) compared to our proposed MM-When2Speak in Table 12. Evident performance gains can be observed compared to text-based Qwen2.5, suggesting including more modalities facilitates the accurate predictions of response types, but our MM-When2Speak still outperforms Qwen2.5-VL and Video-ChatGPT by large margins, despite finetuning.

Table 12: Performance evaluations of response type prediction on Short-Clips-Test dataset for Qwen2.5-VL and MM-When2Speak. P: precision, R: recall. †: Model fintuned on Short-Clips-Train. Our MM-When2Speak is triple modalities.

| Model | Metric | Affm. | Grat. | Fare. | Grtn. | Qstn. | Surp. | Pndr. | FR | Slnc. |
|---|---|---|---|---|---|---|---|---|---|---|
| | | | | | Video + Text | | | | | |
| Qwen2.5-VL | P | 21.89 | 23.87 | 20.32 | 19.69 | 19.30 | 20.88 | 21.19 | 23.78 | 24.52 |
| | R | 17.65 | 19.83 | 17.25 | 17.14 | 16.01 | 15.42 | 16.98 | 20.09 | 19.49 |
| Qwen2.5-VL† | P | 33.52 | 39.14 | 40.96 | 34.82 | 27.43 | 28.38 | 26.64 | 34.52 | 45.19 |
| | R | 30.79 | 32.74 | 36.15 | 35.57 | 23.87 | 25.66 | 27.26 | 38.67 | 40.49 |
| Video-ChatGPT† | P | 37.66 | 33.95 | 40.44 | 30.79 | 33.69 | 32.15 | 32.05 | 40.22 | 40.73 |
| | R | 45.29 | 39.27 | 40.02 | 40.28 | 34.92 | 32.67 | 24.17 | 46.73 | 37.88 |
| MM-When2Speak | P | **62.21** | **64.35** | **63.15** | **63.29** | **46.26** | **50.52** | **37.78** | **68.15** | **68.78** |
| | R | **59.86** | **61.99** | **60.79** | **60.44** | **43.91** | **46.25** | **35.45** | **65.79** | **66.42** |

## D.3 TRIPLE MODALITIES: VITA-1.5 (VISION+SPEECH+TEXT)

We compare our method with the finetuned VITA-1.5, and results are reported in Table 13. We can see that, although our MM-When2Speak is still the best performing method, by adopting all three modalities, VITA-1.5 achieves competitive performance compared to Qwen2.5 and Qwen2.5-VL, and finetuning enables it to achieve further improvements. This indicates that multimodality is crucial to accurately predict response types in dyadic conversations.

Table 13: Performance evaluations of response type prediction on Short-Clips-Test dataset for VITA-1.5 and MM-When2Speak. P: precision, R: recall. †: Model fintuned on Short-Clips-Train. Our MM-When2Speak is triple modalities.

| Method | Metric | Affm. | Grat. | Fare. | Grtn. | Qstn. | Surp. | Pndr. | FR | Slnc. |
|---|---|---|---|---|---|---|---|---|---|---|
| | | | | Video + Speech + Text | | | | | | |
| VITA-1.5 | P | 38.03 | 39.68 | 39.16 | 38.81 | 28.46 | 31.23 | 23.32 | 31.36 | 35.32 |
| | R | 35.14 | 36.79 | 36.27 | 35.92 | 25.61 | 28.37 | 20.50 | 28.50 | 32.44 |
| VITA-1.5† | P | 60.51 | 63.16 | 62.03 | 61.76 | 45.16 | 46.91 | 36.91 | 65.44 | 68.72 |
| | R | 56.83 | 59.47 | 58.64 | 58.07 | 41.51 | 45.56 | 35.11 | 61.75 | 65.01 |
| MM-When2Speak | P | **62.21** | **64.35** | **63.15** | **63.29** | **46.26** | **50.52** | **37.78** | **68.15** | **68.78** |
| | R | **59.86** | **61.99** | **60.79** | **60.44** | **43.91** | **46.25** | **35.45** | **65.79** | **66.42** |

Table 14: Performance comparison between our method and VITA-1.5 with different degradations.

| Model | Degradation | Metric | Affm. | Grat. | Fare. | Grtn. | Qstn. | Surp. | Pndr. | FR | Slnc. |
|---|---|---|---|---|---|---|---|---|---|---|---|
| MM-When-2Speak | - | P | **62.21** | **64.35** | **63.15** | **63.29** | **46.26** | **50.52** | **37.78** | **68.15** | **58.78** |
| | | R | **59.86** | **61.99** | **60.79** | **60.44** | **43.91** | **46.25** | **35.45** | **65.79** | **66.42** |
| | +Audio Noise | P | 58.74 | 61.50 | 60.74 | 62.96 | 45.67 | 48.84 | 37.66 | 64.28 | 55.99 |
| | | R | 55.16 | 57.08 | 59.33 | 55.87 | 40.28 | 42.95 | 34.49 | 62.73 | 64.89 |
| | +Downgraded Images | P | 58.74 | 61.50 | 60.74 | 62.96 | 45.67 | 48.84 | 37.66 | 64.28 | 55.99 |
| | | R | 55.16 | 57.08 | 59.33 | 55.87 | 40.28 | 42.95 | 34.49 | 62.73 | 64.89 |
| VITA-1.5 | - | P | 38.03 | 39.68 | 39.16 | 38.81 | 28.46 | 31.23 | 23.32 | 31.36 | 35.32 |
| | | R | 35.14 | 36.79 | 36.27 | 35.92 | 25.61 | 28.37 | 20.50 | 28.50 | 32.44 |
| | +Audio Noise | P | 31.17 | 30.95 | 32.96 | 33.02 | 22.90 | 27.97 | 22.08 | 26.76 | 32.41 |
| | | R | 28.46 | 35.23 | 28.57 | 26.89 | 24.16 | 21.44 | 16.66 | 23.48 | 30.63 |
| | +Downgraded Visual | P | 31.89 | 28.16 | 31.85 | 33.15 | 23.19 | 26.52 | 20.19 | 25.14 | 33.96 |
| | | R | 29.17 | 33.43 | 26.68 | 25.25 | 20.08 | 26.75 | 19.62 | 21.33 | 28.69 |

Table 15: Cross-domain performance comparison on Behavior-SD dataset Lee et al. (2025) with speech+text modality setting.

| Model | Metric | Affm. | Grat. | Fare. | Grtn. | Qstn. | Surp. | Pndr. | FR | Slnc. |
|---|---|---|---|---|---|---|---|---|---|---|
| ChatGPT | P | 22.49 | 17.71 | 100.00 | 19.60 | 20.88 | 19.67 | 21.06 | 28.16 | 30.66 |
| | R | 19.50 | 16.23 | 100.00 | 18.57 | 17.89 | 17.61 | 23.98 | 24.68 | 27.60 |
| VITA-1.5 | P | 24.15 | 19.71 | 100.00 | 24.25 | 18.33 | 23.76 | 18.63 | 28.86 | 32.04 |
| | R | 21.11 | 13.93 | 100.00 | 22.02 | 16.86 | 20.73 | 20.37 | 26.90 | 28.80 |
| Qwen2-Audio | P | 23.43 | 23.09 | 100.00 | 17.12 | 18.01 | 25.67 | 19.66 | 24.37 | 28.10 |
| | R | 25.87 | 19.74 | 100.00 | 15.10 | 20.37 | 29.59 | 22.22 | 25.01 | 24.51 |
| MM-When2Speak | P | 27.17 | 25.12 | 100.00 | 25.69 | 24.12 | 28.54 | 27.68 | 31.83 | 33.77 |
| | R | 24.88 | 29.91 | 100.00 | 23.79 | 22.38 | 25.90 | 33.02 | 33.26 | 30.19 |

# E    ROBUSTNESS EVALUATION TO NOISY INPUTS

We conduct an experiment to evaluate the robustness of our model to noisy data. We utilize two different noisy inputs: **audio noise**, and **downgraded images**. For audio noise, we add scaled Gaussian noises to the original input audio; for downgraded images, we downsample and then upsample images with a scaling factor of 4. All other modalities are kept untouched during the degradation. Table 14 shows results reported with different degradations to the input from Short-Clips dataset comparing our proposed method and VITA-1.5. From the table, we observe that performance for both models (ours and VITA-1.5) drops when dealing with degraded input quality, but our method obtains a relatively smaller performance degradation compared to VITA-1.5. This suggests that our proposed MM-When2Speak is comparatively robust to noisy inputs.

# F    CROSS EVALUATION ON BEHAVIOR-SD DATASET

We further conduct a cross-domain evaluation on Behavior-SD Lee et al. (2025), **an speech-text full-duplex synthesized dialog dataset**. We organize and sample a total of 6,000 instances from the original dataset, with 2,000 for reaction, `full_response`, and `silence`, respectively. We use the speech+text model from the ablation study in our main paper to conduct evaluations and compare with other speech+text LLMs, as reported in Table 15. We can see that MM-When2Speak still surpasses others by evident margins, which validates that our method is able to generalize to out-domain data.

It is noteworthy that, Behavior-SD is a synthesized dataset and only contains data of speech and text modalities, while our collected dataset is based on conversation videos with visual, audio, and text modalities, which can provide **more cues in capturing subtle conversational dynamics** for response type prediction potentially more suitable for real-world applications.

# G    COMPARISON WITH BACKCHANNEL DETECTION METHOD

In this section, we conduct an experiment to compare our proposed MM-When2Speak with a Transformer-based backchannel detection method One Stream Amer et al. (2023), which is open-source and published, to further evaluate on our Short-Clips reaction dataset, and compare it to our method. We follow and utilize One Stream's official implementations to **recognize backchannel detection as a binary task**, and convert our dataset labels to **reaction** (1, including all reaction data)

Table 16: Backchannel detection performance comparison between One Stream and our MM-When2Speak. The "Multi-class Cls. Recall" denote the recall for each reaction type by dissecting the binary classification results.

| Method | Binary Cls. Accuracy | Multi-class Cls. Recall | | | | | | |
|---|---|---|---|---|---|---|---|---|
| | | Affm. | Grat. | Fare. | Grtn. | Qstn. | Surp. | Pndr. |
| One Stream | 48.32 | 56.33 | 53.63 | 48.17 | 52.09 | 30.66 | 37.12 | 29.68 |
| MM-When2Speak | **58.61** | **59.86** | **61.99** | **60.79** | **60.44** | **43.91** | **46.25** | **35.45** |

and **no reaction** (0, including `full_response` and `silence` data). We divide this converted dataset into training and testing sets following the same 7:3 ratio, and compute the evaluation results following One Stream's evaluation metric (Binary classification accuracy), as reported in Table 16. Additionally, we report Recall for each backchannel reaction class, which is done by analyzing each backchannel detection result and examining its originally corresponding reaction label. It can be observed that our method outperforms One Stream by 10.29% in accuracy for binary classification, and achieves consistently better results in multi-class classification recall, demonstrating the effectiveness of our method for binary backchannel detection.

We also plot the confusion matrix for One Stream and our MM-When2Speak, as in Fig. 12, where we normalize the confusion matrix over all samples and enhance the color contrast to highlight the difference. We can see that MM-When2Speak produces a confusion matrix concentrated on the diagonal with higher true positive and true negative rates, indicating more reliable predictions for both classes.

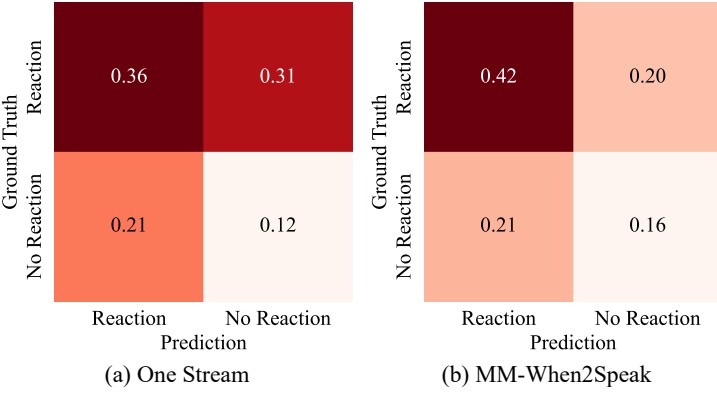

(a) One Stream        (b) MM-When2Speak

Figure 12: Confusion Matrix for binary classification of backchannel reaction.

## H    INTER-ANNOTATOR AGREEMENT ON HUMAN EVALUATIONS

We now provide full agreement statistics and per-class analyses. The testset used for human verification consists of 27 Affirmations, 2 Farewells, 6 Gratitudes, 3 Greetings, 6 Ponderings, 4 Questions, 6 Surprises, 21 Full Responses, and 30 Silences. The information for the nine voluntary participants are reported in Table 17.

### H.1    AUTO VS. MAJORITY-HUMAN AGREEMENT

We report accuracy, macro-level precision/recall/F1, and a bootstrap confidence interval (CI) in Table 18, where **higher values indicate that the labels generated by our automatic annotation pipeline align better with human judgements**. All results exceed 0.8, validating the fair quality of our annotations.

Using the same Majority-Human labels as reference, we further compute per-class precision, recall, and F1 for the automatic pipeline, shown in Table 19. These metrics similarly show high agreement on major classes (`Affm., FR, Slnc.`) and expected variability on lower-support categories. These results corroborate the human-vs-auto analysis already presented in Sec. 4.4 of the main paper.

Table 17: Basic information for participants in the human study.

| Item | Value |
|---|---|
| Number of participants | 9 |
| Experience | 7/9 with prior multimedia labeling; all instructed with the procedure. |
| Language | All fluent in English |
| Training | 10–15 min guideline session + 10 practice items |
| Compensation | Voluntary |
| Ethics | No data distribution or re-identification; local offline use only |
| Tooling | Google Forms / on-site labeling |

Table 18: Auto vs. Majority-Human Agreement.

| | Accuracy | Macro-Precision | Macro-Recall | Macro-F1 | CI (95%) |
|---|---|---|---|---|---|
| Value | 0.876 | 0.881 | 0.851 | 0.865 | [0.834, 0.882] |

Table 19: Per-class Metrics (Auto vs. Majority-Human).

| Metric | Affm. | Grat. | Fare. | Grtn. | Qstn. | Surp. | Pndr. | FR | Slnc. | Macro-Avg |
|---|---|---|---|---|---|---|---|---|---|---|
| P | 0.91 | 0.80 | 0.78 | 0.78 | 0.71 | 0.80 | 0.69 | 0.92 | 0.94 | — |
| R | 0.87 | 0.50 | 0.75 | 0.67 | 0.67 | 0.75 | 0.63 | 0.89 | 0.90 | — |
| F1 | 0.89 | 0.62 | 0.76 | 0.72 | 0.69 | 0.77 | 0.66 | 0.90 | 0.92 | 0.86 |

## H.2 INTER-ANNOTATOR AGREEMENT BY CLASS

Table 20 reports inter-annotator agreement for each class. For each reaction type we convert the labels into a **binary one-vs-rest scheme** (1 = this class, 0 = other), and compute Fleiss' $\kappa$ on the resulting matrix. In practice:

- $\kappa \geq 0.8$ indicates perfect consistency,
- $0.6 \leq \kappa < 0.8$ indicates substantial consistency,
- $0.4 \leq \kappa < 0.6$ indicates moderate consistency,
- $\kappa < 0.4$ indicates insufficient consistency.

Table 20: Inter-annotator agreement by class (Fleiss' $\kappa$).

| | Affm. | Grat. | Fare. | Grtn. | Qstn. | Surp. | Pndr. | FR | Slnc. | Overall |
|---|---|---|---|---|---|---|---|---|---|---|
| $\kappa$ | 0.78 | 0.49 | 0.66 | 0.52 | 0.53 | 0.61 | 0.55 | 0.80 | 0.85 | 0.71 |

No class shows $\kappa < 0.4$, and overall Fleiss' $\kappa \approx 0.71$ indicates substantial agreement, confirming that **the reaction labels are sufficiently reliable to validate our automatic pipeline**.

## H.3 TOP-5 CONFUSIONS AND FAILURE CASES

Using Majority-Human labels as reference, we compute a confusion-frequency matrix between human annotations and automatic outputs. The five most frequent (out of 105 samples) are given in Table 21.

These confusions mainly occur between semantically adjacent or subtle categories, consistent with human disagreement patterns. Most remaining errors arise from weak prosody, micro-expressions, or window-boundary effects:

- **Pondering vs. Surprise:** both have weak prosody and limited lexical cues, making decisions sensitive to micro-expressions and $\tau = 250$ boundary effects.
- **Affirmation vs. Silence:** very short backchannels ("mm", "yeah") may be filtered by ASR or fall near window edges.
- **Farewell / Greeting / Question:** human agreement is already lower ($\kappa \approx 0.49$–$0.61$).

Table 21: Top-5 confusion pairs (percentage of 105 samples).

| True → Pred | Percentage (%) |
|---|---|
| Question → Surprise | 5.8 |
| Pondering → Silence | 3.9 |
| Affirmation → Question | 2.4 |
| Pondering → Affirmation | 2.9 |
| Gratitude → Greeting | 1.9 |

## I    ABLATION ON ADOPTING DIFFERENT QWEN2 LLMS

We adopt Qwen2-7B-base as the LLM in our MM-When2Speak. To evaluate the impact of instruction-based finetuning of LLM, we **substitute** the original LLM in our model with Qwen2-7B-instruct and compare prediction performance on our Short-Clips-Test dataset, following the same training and testing strategies. The results are reported in Table, where performance of using Qwen2-7B-instruct is overall slightly better than Qwen2-7B-base, but not significantly, which suggests that instruction finetuning might bring limited gains in this problem setting.

Table 22: Performance of our MM-When2Speak with different Qwen2-7B LLMs.

| Model | Metrics | Affm. | Grat. | Fare. | Grtn. | Qstn. | Surp. | Pndr. | FR | Slnc. |
|---|---|---|---|---|---|---|---|---|---|---|
| Ours w/ Qwen2-7B-base | P | 62.21 | 64.35 | 63.15 | 63.29 | 46.26 | 50.52 | 37.78 | 68.15 | 68.78 |
| | R | 59.86 | 61.99 | 60.79 | 60.44 | 43.91 | 46.52 | 35.45 | 65.79 | 66.42 |
| Ours w/ Qwen2-7B-instruct | P | 63.29 | 66.98 | 63.78 | 62.19 | 50.76 | 54.69 | 39.02 | 67.83 | 69.95 |
| | R | 59.97 | 63.29 | 62.01 | 58.48 | 48.39 | 47.03 | 35.79 | 65.48 | 68.28 |

## J    DISCUSSIONS

### J.1    MULTI-PARTY CONVERSATIONS

Our work intentionally focuses on **dyadic (user–agent) conversations**, and this is a core design choice rather than a limitation. The goal of MM-When2Speak is **not** to model arbitrary conversational structures, but to enable **fine-grained prediction of when an agent should react in two-person interactions**, which is a setting for practical applications such as customer–agent interactions, video calls, and AI assistants. Multi-party dialogue involves additional tasks (e.g., speaker diarization, tracking, turn allocation), which are different from our scope.

We conduct a light and scope-preserving evaluation on the ICSI multi-party meeting corpus Janin et al. (2003)[2] without retraining our model. We choose the Bdb001 session (a 50-minute long discussion between six different participants), and preprocess it the same way as in our dataset. We assign participant "chan1" as the agent side, and merge the rest as the user side, and evaluate "audio+text only" models. We extract 400 silence, 36 full-response, and 26 reaction clips. Results are reported with the finetuned VITA-1.5 and zero-shot ChatGPT in Table 23.

Table 23: Scope-preserving evaluation on multi-party data.

| Model | Metrics | Silence | Full Response | Reaction |
|---|---|---|---|---|
| ChatGPT | P | 33.43 | 39.01 | 21.54 |
| | R | 25.38 | 50.00 | 30.77 |
| VITA-1.5 | P | 35.25 | 43.66 | 27.14 |
| | R | 28.94 | 41.66 | 25.03 |
| Ours | P | 41.20 | 51.69 | 35.77 |
| | R | 29.67 | 58.30 | 46.15 |

Despite an unseen setting, our method shows consistently better performance compared to others. This demonstrates that, while our paper intentionally targets dyadic timing, the model can generalize

---

[2]Dataset link: https://groups.inf.ed.ac.uk/ami/icsi/

to the multi-speaker scenario, and multi-party conversational agents will be our **targeted future work** to address more complex conversational scenarios.

### J.2   EXTENSION TO "HOW TO SPEAK"

It is noteworthy that our work is designed to **decouple fine-grained timing/type response prediction from response content generation**. This separation is deliberate: "when to speak" is a **distinct, lower-level control signal** that governs whether a conversational agent should react and what kind of reaction (e.g., affirmation, pondering), while "how/what to speak" belongs to a **different stage** handled by a generator (template, retrieval, or LLM). This modular design reduces system complexity, avoids entangling two heterogeneous problems, and enables clear, measurable evaluation, which is a setting for practical applications such as customer–agent interactions, video calls, and AI assistants.

Integrating response generation lies outside the scope of this work rather than being a limitation. Two straightforward extensions exist:

- **Plug-in generation**: The output of MM-When2Speak can directly serve as a **triggering and type-conditioning signal** for any off-the-shelf generator (LLM APIs, retrieval/template modules, etc.). This provides structured control and avoids unnecessary dense generation.
- **Unified training**: Since our architecture already uses Qwen, which is pretrained with full generation capability, the model could be extended by jointly optimizing **reaction type + response content**, enabling end-to-end generation when needed.

These extensions expand the system-level scope but do not alter the core contribution of this paper, which is **multimodal when-to-speak prediction**. We view generation integration as natural future work rather than a missing component of the current task.

Although generation is beyond the scope of this work, the underlying LLM used in our architecture (Qwen2-7B-base) is **inherently capable of generation**. To illustrate the practical value of predicting response type, we conduct a small proof-of-concept qualitative test using two slightly-modified prompts (only change the control of outputs) to: 1) predict reaction type first, and then generate content based on it; **2)** directly generate a response if necessary without type prediction. We then use these two prompts on our model to qualitatively compare the differences. To ensure generalizability, we use Qwen2-7B-instruct as an alternative in our MM-When2Speak for comprehensive comparison. The qualitative results are reported in Table

Table 24: Qualitative comparisons using Prompt-1 between Qwen2-7B-base and Qwen2-7B-instruct

| Ground Truth Reaction Type | Qwen2-7B-base | Qwen2-7B-instruct |
|---|---|---|
| Affirmation | Affirmation: "Yeah, yeah." | Affirmation: "Yes." |
| Question | Question: "Are you serious?" | Question: "Do you really mean that?" |
| Pondering | Pondering: "Hmm, okay." | Pondering: "Umm well." |

Table 25: Qualitative comparisons using Prompt-2 between Qwen2-7B-base and Qwen2-7B-instruct

| Ground Truth Reaction Type | Qwen2-7B-base | Qwen2-7B-instruct |
|---|---|---|
| Affirmation | "Yes, I totally agree with what you are saying right here. Your comment towards this incoming external threat makes a lot of sense, especially to our country at this stage at this time. And I believe in this scenario we have to have the dominance…" | "Indeed, I completely concur with your point. Your remarks about this emerging external threat are very reasonable, particularly given our nation's current circumstances. Honestly, this situation calls for us to maintain a position of strength and dominance." |
| Question | "I mean, really? I cannot believe it. This is so incredible." | "How could anyone do this? Are you sure about it?" |
| Pondering | "Wow, okay, sure!" | "Hmm, I see what you mean, yeah." |

From the above results we can see that, using Prompt-1, the model **consistently produces both the correct reaction type and appropriate short reaction content**. Without type prediction

as the first step (Prompt-2), the LLM frequently shifts to **irrelevant or full-response behaviors**, demonstrating instability. These observations indicate that our model **can be extended to "how to speak"**, and reaction-type prediction provides **a crucial control signal** for generation, reinforcing the importance of the task addressed in this work, while generation itself remains outside the targeted scope. Additionally, using the instruction-tuned LLM Qwen2-7B-instruct can somewhat slightly improve the output quality, but **not yet significantly**. Although using Qwen2-7B-instruct may lead to potential benefits in terms of an integrated "when and how to speak" system, it is out of the scope of the main contributions of our work, i.e., using multimodal learning to benchmark and improve the "when to speak" task.

### J.3 DIFFERENCES TO TEMPORAL/EVENT-BOUNDARY METHODS

Temporal/event-boundary models, such as causal streaming, CTC-style detector architectures, excel in **frame-level event segmentation** where boundaries are sharp and unambiguous (e.g., phoneme, word, or VAD decisions). These models are designed to decide *exactly* when a discrete symbol starts or ends.

In contrast, our task is **fundamentally semantic and multi-class**, with **inherently fuzzy reaction boundaries**. A reaction (e.g., pondering, surprise, affirmation) does not correspond to a precise frame-level onset; annotations are therefore **window-level with temporal tolerance**, not instantaneous events. Modeling must integrate multimodal cues (i.e., facial motion, prosody, lexical content) over a meaningful temporal context to output **one coherent reaction type**.

Our architecture is thus **task-aligned**: a tri-modal fusion encoder operating on a 10-second context is a natural fit for window-level semantic labels and enables clean integration into real-time sliding-window inference without altering the label space or supervision format. Adopting boundary-focused models would require **redefining labels at frame resolution**, handling overlapping or ambiguous reactions, and adding nontrivial latency and engineering complexity, while the evaluation remains window-based. We therefore view CTC/boundary architectures as **different-scope extensions**, not direct baselines for the current formulation.

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
