# OpenReview forum: "Beyond Words: Multimodal LLM Knows When to Speak"
_ICLR.cc/2026/Conference — Submitted to ICLR 2026_

### Official Review · Reviewer_NMqH · 2025-10-27

**Soundness:** 3
**Presentation:** 3
**Contribution:** 3
**Rating:** 6
**Confidence:** 4

**Summary:**

This paper argues a critical limitation of large language model (LLM)-based chatbots: their inability to accurately determine "when to speak" (especially for brief, reactive utterances) in real-world conversations, which stems from overreliance on single-modal inputs (lacking visual and auditory cues). To solve this, the research proposes a multimodal framework and corresponding dataset, with key contributions as follows: 1) A Multimodal Conversational Dataset. 2) A Multimodal LLM for “when to speak” Prediction. The experimental results show that adding modalities consistently improves accuracy, and the MM-When2Speak outperforms baselines.

**Strengths:**

1. The paper addresses a key limitation of existing LLM-based chatbots—their inability to accurately determine "when to speak" (especially for brief, real-time backchannel reactions) in natural conversations—rather than just "what to say." This focus fills a critical gap: while most research prioritizes response content coherence, the timing and appropriateness of responses are equally vital for human-like interaction. By tackling this underexplored challenge, the work directly enhances the practicality of conversational AI in real-world scenarios.
2. The paper constructs a high-quality, multimodal dataset with fine-grained annotations. A major strength lies in the creation of a curated, multimodal dataset that addresses the scarcity of resources integrating visual, auditory, and textual cues for response timing prediction.
3. The paper proposes a well-designed multimodal model with adaptive fusion. The model surpasses baselines on the "when to speak" prediction task.

**Weaknesses:**

1. The dataset, while carefully curated, has notable limitations in scope that may restrict the generalizability of the findings. It is sourced exclusively from public YouTube videos, focusing on only two conversational scenarios: virtual Zoom meetings and broadcast news interviews (e.g., CNN interviews) . This narrow scenario coverage fails to include other common real-world dialogue contexts.
2. The dataset is limited to interactions between two participants , with no exploration of multi-party conversations.
3. The dataset is relatively small compared to large-scale multimodal datasets in related fields.
4. The paper simplifies the conversational dynamics. The paper’s framing of "when to speak" as a dense classification task using fixed 10-second sliding windows (with 0.5-second strides)  oversimplifies the complexity of natural conversational dynamics. Human dialogue is highly variable in pace.

**Questions:**

1. Do you recognise the limitations of the dataset? Do you have plans to further improve the dataset?
2. MM-When2Speak uses a fixed 10-second sliding window for prediction, but human dialogue often has variable turn lengths (e.g., 2-second backchannels vs. 20-second explanations). Have you experimented with adaptive window sizes (e.g., windows that shrink/grow based on speech pause duration) or dynamic stride lengths?
3. For the three-modal (Video+Speech+Text) baseline. Why did you not include other well-established multimodal LLMs like Video-ChatGPT (Maaz et al., 2024)?

---

> ### Author Response · Authors · 2025-11-22
> **Responses to Reviewer NMqH (1/2)**
>
> We appreciate the reviewer’s thoughtful feedback. Below, we address the reviewer’s feedback and will incorporate our answers in the revised paper.
>
> ## Q1: Scale and generalizability to complex scenarios
>
> ### 1. Problem/Data limitation
>
> Thanks for raising this point. Our work intentionally focuses on **dyadic (user–agent) conversations**, and this is a core design choice rather than a limitation. As stated in the main paper (L32–41) and the supplementary material (L24-30), the goal of MM-When2Speak is **not** to model arbitrary conversational structures, but to enable **fine-grained prediction of *when an agent should react* in two-person interactions**, which is a setting for practical applications such as customer–agent interactions, video calls, and AI assistants.
>
> Integrating response generation lies outside the scope of this work rather than being a limitation. Two straightforward extensions exist:
>
> - **Plug-in generation**: The output of MM-When2Speak can directly serve as a **triggering and type-conditioning signal** for any off-the-shelf generator (LLM APIs, retrieval/template modules, etc.). This provides structured control and avoids unnecessary dense generation.
> - **Unified training**: Since our architecture already uses **Qwen2**, which is pretrained with full generation capability, the model could be extended by jointly optimizing **reaction type + response content**, enabling end-to-end generation when needed.
>
> These extensions expand the system-level scope but do not alter the core contribution of this paper, which is **multimodal when-to-speak prediction**. We view generation integration as natural future work rather than a missing component of the current task.
>
> ### 2. Generalization to multi-party scenarios
>
> Although multi-party dialogue involves additional tasks (e.g., speaker diarization, tracking, turn allocation) and is different from our scope, we still conduct a light and scope-preserving evaluation on the ICSI multi-party meeting corpus [1] without re-training our model. We choose the Bdb001 session (a 50-minute long discussion between six different participants), and preprocess it the same way as in our dataset. We assign participant "chan1" as the agent side, and merge the rest as the user side, and evaluate "audio+text only" models. We extract 400 silence, 36 full-response, and 26 reaction clips. Results (P/R in %) are reported with the finetuned VITA-1.5 and zero-shot ChatGPT:
>
> | Model | Silence | Full Response | Reaction |
> | --- | --- | --- | --- |
> | ChatGPT | 33.43 / 25.38 | 39.01 / 50.00 | 21.54 / 30.77 |
> | VITA-1.5 | 35.25 / 28.94 | 43.66 / 41.66 | 27.14 / 25.03 |
> | Ours | 41.20 / 29.67 | 51.69 / 58.30 | 35.77 / 46.15 |
>
> Despite an unseen setting, our method shows **consistently better performance** compared to others. This demonstrates that, while our paper intentionally targets dyadic timing, the model can generalize appropriately as multi-speaker preprocessing is available.
>
> [1] ICSI corpus. (n.d.). https://groups.inf.ed.ac.uk/ami/icsi/

---

> > ### Comment · Reviewer_NMqH · 2025-11-26
> > **maintain the rating**
> >
> > I thank the authors for their detailed response, and I still have concerns about the limitations of the user–agent conversations setting. I will maintain my original rating.

---

> > > ### Author Response · Authors · 2025-11-30
> > > **Response to reviewer NMqH "maintain the rating" on 26 Nov 2025, 04:11**
> > >
> > > We thank the reviewer again for carefully revisiting our rebuttal. We acknowledge that restricting to dyadic user–agent interactions is different in terms of the most general conversational settings, and we respect the reviewer’s perspective on this point. Our intention in this paper, however, is to take a **clean and controllable first step** toward modeling multimodal response timing/type prediction, and the dyadic setting was **chosen deliberately to isolate core interaction dynamics** (reaction timing, turn-taking, responsiveness) without conflating them with additional modules (e.g., speaker diarization, tracking, or turn allocation).
> > >
> > > At the same time, we have made a conscious effort in the rebuttal (responses of Q1) to go beyond a purely dyadic setup and to demonstrate that **our method is not bounded to the original data**. Concretely, we:
> > >
> > > - Clarified how our model can be plugged into downstream generators (e.g., LLM) and how unified training of "when to speak" and "how to speak" could be realized on top of the same backbone;
> > > - **Conducted a multi-party generalization test on the ICSI meeting corpus**, where we treat one channel as the agent and merge the remaining speakers as the "user side". Without retraining the model, our method achieves better P/R than both finetuned VITA-1.5 and zero-shot ChatGPT on Silence / Full Response / Reaction categories in this six-speaker, 50-minute meeting, which indicates that the learned representations **can transfer reasonably to more complex multi-speaker scenarios**.
> > >
> > > We will revise the paper to make these points clearer: stating the dyadic user–agent focus as an intentional design choice, explicitly acknowledging its limitations for fully general multi-party interaction, and summarizing both the ICSI experiment and discussions toward richer conversational structures. We sincerely appreciate the reviewer’s thoughtful feedback and understand the decision to maintain the originally positive rating.

---

> ### Author Response · Authors · 2025-11-22
> **Responses to Reviewer NMqH (2/2)**
>
> ## Q2: Performance using different window sizes / strides
>
> We report the ablation study here regarding the size/stride of sliding window on Full-Videos dataset. The following first table ablates the window size (size) using a default stride of 0.5s, and the second table varies the stride with default window size 10s. **Notably the performance peaks when size=10s and stride=0.5s**.
>
> **Table 1: Performance on different window sizes (stride=0.5s)**
>
> |  | size=1s | size=2s | size=5s | size=10s | size=15s | size=20s |
> | --- | --- | --- | --- | --- | --- | --- |
> | Affm. | 27.42 / 25.11 | 28.14 / 26.42 | 30.44 / 28.71 | 31.55 / 29.24 | 29.41 / 27.55 | 27.95 / 26.17 |
> | Grat. | 28.36 / 26.12 | 29.57 / 27.14 | 31.44 / 28.94 | 32.25 / 29.95 | 30.52 / 28.44 | 28.78 / 26.91 |
> | Fare. | 27.15 / 25.02 | 28.33 / 26.11 | 30.66 / 27.55 | 31.25 / 28.94 | 29.14 / 27.01 | 27.66 / 25.55 |
> | Grtn. | 34.41 / 22.14 | 36.22 / 24.88 | 38.11 / 25.94 | 39.53 / 27.22 | 37.08 / 24.66 | 35.22 / 23.55 |
> | Qstn. | 24.55 / 22.14 | 25.91 / 23.57 | 27.41 / 24.88 | 28.21 / 25.91 | 26.55 / 24.12 | 25.41 / 23.01 |
> | Surp. | 25.71 / 23.11 | 26.84 / 24.02 | 27.91 / 25.77 | 28.82 / 26.52 | 27.14 / 25.01 | 25.94 / 23.72 |
> | Pndr. | 24.02 / 22.31 | 25.11 / 23.88 | 26.71 / 24.72 | 27.47 / 25.17 | 26.04 / 24.22 | 24.91 / 23.11 |
> | FR    | 31.02 / 28.14 | 32.14 / 29.01 | 33.77 / 30.72 | 35.17 / 32.85 | 33.66 / 31.44 | 31.84 / 29.77 |
> | Slnc. | 30.12 / 27.11 | 31.44 / 28.63 | 32.55 / 29.66 | 33.27 / 30.95 | 32.22 / 29.40 | 30.47 / 28.52 |
>
> **Table 2: Performance on different window strides (window size=10s)**
>
> |  | stride=0.1 | stride=0.2 | stride=0.5 | stride=1 | stride=2 | stride=5 |
> | --- | --- | --- | --- | --- | --- | --- |
> | Affm. | 28.15 / 25.83 | 29.76 / 27.34 | 31.55 / 29.24 | 29.59 / 27.09 | 27.28 / 25.72 | 25.45 / 24.08 |
> | Grat. | 29.24 / 26.36 | 30.98 / 27.52 | 32.25 / 29.95 | 29.73 / 27.97 | 27.73 / 26.59 | 26.25 / 23.66 |
> | Fare. | 28.18 / 25.86 | 29.88 / 26.91 | 31.25 / 28.94 | 29.00 / 26.63 | 27.87 / 25.57 | 24.84 / 23.39 |
> | Grtn. | 34.96 / 24.51 | 36.66 / 25.05 | 39.53 / 27.22 | 37.18 / 25.59 | 35.31 / 23.69 | 30.96 / 21.72 |
> | Qstn. | 24.74 / 23.23 | 26.25 / 24.80 | 28.21 / 25.91 | 25.88 / 24.20 | 25.10 / 22.93 | 22.10 / 20.19 |
> | Surp. | 25.34 / 23.29 | 26.64 / 25.04 | 28.82 / 26.52 | 26.84 / 25.19 | 25.47 / 23.28 | 22.44 / 21.59 |
> | Pndr. | 24.62 / 23.11 | 26.02 / 23.48 | 27.47 / 25.17 | 25.87 / 23.11 | 24.65 / 22.77 | 21.46 / 19.78 |
> | FR    | 31.13 / 29.18 | 33.13 / 30.27 | 35.17 / 32.85 | 33.38 / 30.02 | 30.91 / 29.14 | 27.98 / 26.80 |
> | Slnc. | 30.12 / 28.48 | 30.66 / 28.68 | 33.27 / 30.95 | 30.83 / 29.14 | 28.66 / 27.52 | 26.03 / 24.69 |
>
> We will include these results in our revised paper.
>
> ## Q3: Dataset extension
>
> We acknowledge that the current dataset has some limitations in terms of scale and complexity. However, to the best of our knowledge, our work is among the first to explicitly address the “when to speak” task. Importantly, we focus on **curating a high-quality dataset and establishing a rigorous baseline study and evaluation protocol**, providing a strong foundation for future research. In future work, we plan to further enhance the dataset by collecting more balanced samples for rare reaction types, incorporating more diverse speakers and domains, and exploring finer-grained temporal annotations or auxiliary signals (e.g., micro-expression markers or prosodic change points).
>
> ## Q4: Comparison to Video-ChatGPT
>
> We conduct experiments on the Short-Clips dataset using our method and Video-ChatGPT, and the following table reports the performance comparison. Although Video-ChatGPT performs slightly better on Affirmation and Pondering to our method on the "text+vision" modalities, **our method still achieves evidently better results in most types**. In addition, our method using "text+vision+speech" modalities achieves the best results, validating the effectiveness of using multimodality in solving this problem.
>
> |  | Video-ChatGPT | Ours (Text+Vision) | Ours (Text+Vision+Speech) |
> | --- | --- | --- | --- |
> | Affm. | 37.66 / 45.29  | 36.89 / 44.22  | 62.21 / 59.86 |
> | Grat. | 33.95 / 39.27  | 38.65 / 44.41  | 64.35 / 61.99 |
> | Fare. | 40.44 / 40.02  | 41.09 / 40.45  | 63.15 / 60.79 |
> | Grtn. | 30.79 / 40.28  | 38.83 / 42.93  | 63.29 / 60.44 |
> | Qstn. | 33.69 / 34.92  | 34.22 / 36.07  | 46.26 / 43.91 |
> | Surp. | 32.15 / 32.67  | 32.71 / 33.72  | 50.52 / 46.25 |
> | Pndr. | 32.05 / 24.17  | 31.75 / 24.12  | 37.78 / 35.45 |
> | FR    | 40.22 / 46.73  | 43.49 / 46.05  | 68.15 / 65.79 |
> | Slnc. | 40.73 / 37.88  | 42.32 / 41.49  | 68.78 / 66.42 |

---

### Official Review · Reviewer_7wEs · 2025-10-30

**Soundness:** 3
**Presentation:** 2
**Contribution:** 2
**Rating:** 6
**Confidence:** 3

**Summary:**

This paper addresses real-time conversational response planning. The authors build a multimodal model that predicts when a system should speak, stay silent, or give short listener reactions. They compile a new dataset of natural two-party conversations, and train a sliding-window classifier that takes speech, video, and transcript as input. Experiments show that the proposed model outperforms several multimodal and audio-visual LLM baselines.

**Strengths:**

1. The problem is practical and underexplored, most dialogue systems focus on what to say, but rarely model when they should respond or remain silent.

2. The dataset is a useful contribution: multimodal, time-aligned, and supported by high-quality human-verified annotations.

3. The method is straightforward and computationally lightweight, making real-time deployment feasible.

4. Experimental results indicate clear improvements over strong multimodal LLM baselines across multiple settings.

**Weaknesses:**

1. One concern is the practical usefulness of the proposed model in real applications. Since it only outputs a response type rather than the actual content, it seems more like an auxiliary module that provides a triggering signal for another system to generate the verbal response. The paper does not discuss how this module integrates with a full conversational pipeline.

2. The evaluation does not report latency or real-time efficiency. Given that the model is intended for online interaction, end-to-end latency is critical, and it also relates to the concern above about practicality.

3. The model is trained and tested primarily on the authors' own dataset, which mainly consists of two-party, news/interview-style conversations. It remains unclear how well it would generalize to casual, noisy, or multi-speaker settings.

4. The pre-training stage is mentioned but not clearly described. It is not obvious what exact objective is optimized, how multimodal alignment is learned, and to what extent this pre-training contributes to downstream performance.

**Questions:**

1. How would this module be integrated with a downstream response generation system in a real conversational agent?

2. What is the actual runtime latency under streaming input? Can the model maintain real-time performance in longer conversations?

3. Do the authors have any evidence of generalization beyond the curated dataset, such as informal chats, noisy environments, or multi-speaker scenarios?

4. Can the authors provide more details about the pre-training stage?

**Details Of Ethics Concerns:**

No ethics concerns.

---

> ### Author Response · Authors · 2025-11-22
> **Responses to Reviewer 7wEs (1/2)**
>
> We appreciate the reviewer’s thoughtful feedback. Below, we address the reviewer’s feedback and will incorporate our answers in the revised paper.
>
> ## W1: Method practicality in real applications
>
> Thanks for raising this point. As described in our paper and supplementary material, our work explicitly targets **dyadic (user–agent) conversations** in applications such as two-person video calling, aiming to enable conversational agents to produce context-aware, timely utterances. Extending to multi-party or other complex settings is feasible but would require additional components (e.g., speaker diarization, tracking, and turn allocation), which are different to our core contribution. Moreover, two-person interactions offer a relatively **more controlled and well-defined configuration for learning essential interaction dynamics** (e.g., reaction, turn-taking, responsiveness, behavioral coordination). We therefore consider the dyadic setting **a necessary first step** toward modeling more complex scenarios; however, we provide more results and discussions in W3.
>
> Although equipping response generation capability to our model is beyond the scope of this work, one can **plug in an off-the-shelf generation LLM API for generating response content**. Moreover, the prediction result from our model can be used as **a triggering signal** that not only indicates whether to say anything or not, but also what specific type of response should be generated. This mechanism can potentially **reduce complexity for dense generation**. Second, since our model utilizes Qwen2 as our LLM, which has **inherent generation capability**, one can **tweak the training objective with both response type and the corresponding content** in order to make our model also generate response if necessary. Therefore, we believe our model can be extended to bigger scopes without introducing great challenges, and the above two methods can be our future work.
>
> ## W2: Latency evaluation
>
> We provide latency measurements on the Short-Clips-Test dataset using the same hardware as the paper (NVIDIA L40s). Results (in seconds) are shown below:
>
> | Model | Modality | Average | Min. | Max. |
> | --- | --- | --- | --- | --- |
> | Qwen-2.5 | Text | 0.103 | 0.068 | 0.388 |
> | Ours | Text | 0.092 | 0.073 | 0.369 |
> | Qwen2.5-VL | Text+Vision | 0.561 | 0.217 | 0.835 |
> | Ours | Text+Vision | 0.606 | 0.235 | 0.825 |
> | VITA-1.5 | Text+Vision+Speech | 1.132 | 0.779 | 1.889 |
> | Ours | Text+Vision+Speech | 1.145 | 0.698 | 1.732 |
>
> Latency naturally increases with more modalities. Importantly, under **each modality configuration**, our model exhibits **comparable or lower latency** relative to baselines, despite producing stronger accuracy. This demonstrates that MM-When2Speak remains practically feasible for real-time deployment.
>
> ## W3: Model generalization to degraded data and multi-party scenarios
>
> We have reported results to **degraded audio and vision inputs in Sec. 5 of the supplementary material**, and the results show that our model achieves relatively consistent performance without any significant drops.
>
> As we mentioned in W1, our method is not targing **multi-party conversational scenarios**. To still address the reviewer’s feedback, we conduct a light and scope-preserving evaluation on the ICSI multi-party meeting corpus [1] without re-training our model. We choose the Bdb001 session (a 50-minute long discussion between six different participants), and preprocess it the same way as in our dataset. We assign participant "chan1" as the agent side, and merge the rest as the user side, and evaluate "audio+text only" models. We extract 400 silence, 36 full-response, and 26 reaction clips. Results (P/R in %) are reported with the finetuned VITA-1.5 and zero-shot ChatGPT:
>
> | Model | Silence | Full Response | Reaction |
> | --- | --- | --- | --- |
> | ChatGPT | 33.43 / 25.38 | 39.01 / 50.00 | 21.54 / 30.77 |
> | VITA-1.5 | 35.25 / 28.94 | 43.66 / 41.66 | 27.14 / 25.03 |
> | Ours | 41.20 / 29.67 | 51.69 / 58.30 | 35.77 / 46.15 |
>
> Despite an unseen setting, our method shows **consistently better performance** compared to others. This demonstrates that, while our paper intentionally targets dyadic timing, the model can generalize appropriately for the multi-speaker scenario.
>
> [1] ICSI corpus. (n.d.). https://groups.inf.ed.ac.uk/ami/icsi/

---

> ### Author Response · Authors · 2025-11-22
> **Responses to Reviewer 7wEs (2/2)**
>
> ## W4: Pretraining strategy
>
> Refrencing VITA-1.5, a frozen vision encoder (InternViT-300M) and a frozen audio encoder (Freeze-Omni) are used to extract frame-level visual features and short-segment acoustic features, respectively. These features are passed through lightweight projection layers to map them into the token embedding space of a causal language model backbone (Qwen2-7B). The projected visual and audio tokens are then concatenated with the textual tokens and fed into the LLM in a standard left-to-right fashion.
>
> Pre-training optimizes a **single auto-regressive language-modeling objective**. Given the multimodal context (images/video frames + speech segments + textual prompts), the model predicts the next text token. **Multimodal alignment is learned implicitly** because the LLM must attend to the projected vision and audio tokens in order to minimize the next-token prediction loss on the multimodal conversational and instruction-style data. During this stage, **the LLM and projection layers are trained**, while the vision/audio encoders are **kept frozen or only lightly fine-tuned**.
>
> This pre-training **conditions the backbone LLM with multimodal capability**. It learns to treat visual frames and speech features as context tokens on equal footing with text. As a result, downstream tasks like our real-time multimodal prediction can be handled with lightweight task-specific heads and fine-tuning, because the core model has already learned a **shared latent space and attention patterns** that integrate visual, acoustic, and textual cues.

---

### Official Review · Reviewer_vS6n · 2025-10-31

**Soundness:** 3
**Presentation:** 3
**Contribution:** 2
**Rating:** 2
**Confidence:** 4

**Summary:**

The paper tackles “when should an LLM speak, and with what short reaction?” in dyadic video conversations. It builds two datasets (Short-Clips and Full-Videos) from real conversational videos with aligned video, audio, and text, and proposes MM-When2Speak, a multimodal LLM-based classifier that fuses the three modalities and uses a sliding window for online inference. Experiments against strong text-only and multimodal LLM baselines (e.g., GPT-4o, Gemini-1.5, VITA-1.5) report notable gains, including up to 4× improvement in response-timing accuracy over text-only LLMs.

**Strengths:**

Targeting the underexplored yet vital problem of “when to speak” (vs. “what to say”) in human–AI interaction, the paper reports consistent gains across datasets, modalities, and strong baselines with informative ablations, adopts sliding-window inference for online deployability, and documents transparent data construction and labeling (including confusion matrices and class-wise recall).

**Weaknesses:**

1. The pipeline figure is ambiguous—window length/stride and label–time alignment are unspecified, the transcript appears stretched across multiple audio windows, and it’s unclear whether outputs are per-window predictions or post-processed events.
2. Lacks qualitative case analyses and a breakdown of reaction/query diversity (per-class distribution, long-tail behavior, representative successes/failures).
3. No end-to-end latency or memory profiling in realistic settings; please quantify per 10-s clip on commodity CPU/GPU and compare against lighter baselines to demonstrate real-time feasibility.
4. The architecture is a fairly straightforward tri-modal fusion atop an LLM; motivate why this design is preferable to stronger temporal/event-boundary models (e.g., causal streaming, CTC-style detectors) and provide deeper modeling insights.
5. Auto-labeling with limited human checks needs stronger evidence—report inter-annotator agreement (e.g., Cohen’s κ/α) by class and timing tolerance, plus failure analyses for nuanced reactions (e.g., *pondering* vs *surprise*).
6. Comparisons to GPT-4o/Gemini may be prompt- and seed-sensitive; include prompt ablations, few-shot variants, and variance across runs/seeds.

**Questions:**

1. How do different window lengths/strides affect early vs. late backchannel detection?
2. What fraction of the dataset received human verification?

---

> ### Author Response · Authors · 2025-11-22
> **Responses to Reviewer vS6n (1/4)**
>
> We appreciate the reviewer’s thoughtful feedback, and we address the reviewer’s concerns as follows.
>
> ## W1: Figure ambiguity
>
> We clarify that the model produces **per-window predictions with no post-processing**. The label of each 10s-window is determined using the temporal rule described in **Sec. 1.3.2 of the supplementary material**:
>
> - A window ending at time $t$ is assigned a response label if $t$ falls within the $\tau=250$ ms temporal neighborhood of that annotated event, i.e., $[t-\tau, t+\tau)$.
> - If no annotation falls within this range, the window is labeled as **silence**.
>
> For transcripts, audio diarization provides word-level timestamps. A word is included in a window if its **starting timestamp** falls within the window’s $[t-\Delta t, t)$ ($\Delta t$ is window size) interval, regardless of whether overlap occurs. This ensures strict temporal consistency across video, audio, and text streams. We will refine the figure and caption to make these alignment rules clearer in the paper.
>
> ## W2: Diversity and case analysis
>
> The per-class distributions are already reported in **Sec. 3.2 of the main paper** and **Sec. 1.3 of the supplementary material**. As noted there, natural conversational reactions are **inherently imbalanced**: high-frequency categories (e.g., Silence, Full Response) dominate real interactions, while nuanced reactions (e.g., Pondering) and boundary events (e.g., Farewell) form the **natural long tail**. For instance, in Short-Clips, Affirmation and Gratitude account for **55.39%** and **18.66%**, whereas Question and Pondering are **<5%**, and Surprise/Greeting/Farewell are **5–7%**. Such skew is expected in open-domain human dialogue and correlates with lower human–human agreement.
>
> Regarding case characteristics, our qualitative analysis shows:
>
> - **Successful predictions tend to occur when:**
>     - **Modalities reinforce each other** (prosody, lexical cues, facial expressions align), e.g., positive tone + smile → Affirmation.
>     - **Visual cues are salient**, e.g., widened eyes or abrupt mouth movement → Surprise/Question.
>     - **Clear transition cues exist in speech**, e.g., explicit query or turn boundary → Full Response.
> - **Failure cases generally arise when:**
>     - **Signals are weak or ambiguous**, such as minimal facial change or monotone speech, often leading the model to default to Silence.
>     - **Modalities contradict**, e.g., smiling during sarcasm, causing confusion between Affirmation and Surprise/Question.
>
> We will include a brief summary of these observations in the final version to clarify dataset diversity and model behavior.
>
> ## W3: Latency profiling
>
> We provide latency measurements on the Short-Clips-Test dataset using the same hardware as the paper (NVIDIA L40s). Results (in seconds) are shown below:
>
> | Model | Modality | Average | Min. | Max. |
> | --- | --- | --- | --- | --- |
> | Qwen-2.5 | Text | 0.103 | 0.068 | 0.388 |
> | Ours | Text | 0.092 | 0.073 | 0.369 |
> | Qwen2.5-VL | Text+Vision | 0.561 | 0.217 | 0.835 |
> | Ours | Text+Vision | 0.606 | 0.235 | 0.825 |
> | VITA-1.5 | Text+Vision+Speech | 1.132 | 0.779 | 1.889 |
> | Ours | Text+Vision+Speech | 1.145 | 0.698 | 1.732 |
>
> Latency naturally increases with more modalities. Importantly, under **each modality configuration**, our model exhibits **comparable or lower latency** relative to baselines, despite producing stronger accuracy. This demonstrates that MM-When2Speak remains practically feasible for real-time deployment.
>
> ## W4: Motivate the differences to temporal/event-boundary models
>
> Thanks for raising this point. CTC-style boundary detectors and causal streaming architectures excel in **frame-level event segmentation** where boundaries are sharp and unambiguous (e.g., phoneme, word, or VAD decisions). These models are designed to decide *exactly* when a discrete symbol starts or ends.
>
> In contrast, our task is **fundamentally semantic and multi-class**, with **inherently fuzzy reaction boundaries**. A reaction (e.g., pondering, surprise, affirmation) does not correspond to a precise frame-level onset; annotations are therefore **window-level with temporal tolerance**, not instantaneous events. Modeling must integrate multimodal cues (i.e., facial motion, prosody, lexical content) over a meaningful temporal context to output **one coherent reaction type**.
>
> Our architecture is thus **task-aligned**: a tri-modal fusion encoder operating on a 10-second context is a natural fit for window-level semantic labels and enables clean integration into real-time sliding-window inference without altering the label space or supervision format.
>
> Adopting boundary-focused models would require **redefining labels at frame resolution**, handling overlapping or ambiguous reactions, and adding nontrivial latency and engineering complexity, while the evaluation remains window-based. We therefore view CTC/boundary architectures as **different-scope extensions**, not direct baselines for the current formulation.

---

> ### Author Response · Authors · 2025-11-22
> **Responses to Reviewer vS6n (2/4)**
>
> ## W5: Evidence for auto-labeling
>
> We now provide full agreement statistics and per-class analyses. Note that the testset we use for human verification is composed of data of 27 Affirmations, 2 farewells, 6 gratitudes, 3 greetings, 6 ponderings, 4 questions, 6 surprises, 21 full responses, and 30 silences.
>
> ### 1. Auto vs Majority-Human Agreement
>
> We include accuracy, macro-level P/R/F1, and a bootstrap confidence interval (CI), where **higher values indicate that the labels generated by our automatic annotation pipeline aligns better with human judgements**. The results are all over 0.8, which validates the fair quality of our annotations.
>
> **Table 1: Auto vs Majority-Human (105 samples)**
>
> | Metric | Value |
> | --- | --- |
> | Accuracy | 0.876 |
> | Macro-Precision | 0.881 |
> | Macro-Recall | 0.851 |
> | Macro-F1 | 0.865 |
> | Bootstrap 95% CI | [0.834, 0.882] |
>
> Using the same Majority-Human labels as reference, we further evaluate **per-class Precision/Recall/F1** for the automatic pipeline in the following Table 2. Per-class metrics similarly show high agreement on major classes (Affirmation, Full Response, Silence) and expected variability on nuanced, low-support categories:
>
> **Table 2: Per-class Metrics (Auto vs Majority-Human)**
>
> | Class | Precision | Recall | F1 |
> | --- | ---: | ---: | ---: |
> | Affm. | 0.91 | 0.87 | 0.89 |
> | Fare. | 0.80 | 0.50 | 0.62 |
> | Grat. | 0.78 | 0.75 | 0.76 |
> | Grtn. | 0.78 | 0.67 | 0.72 |
> | Pndr. | 0.71 | 0.67 | 0.69 |
> | Qstn. | 0.80 | 0.75 | 0.77 |
> | Surp. | 0.69 | 0.63 | 0.66 |
> | FR | 0.92 | 0.89 | 0.90 |
> | Slnc. | 0.94 | 0.90 | 0.92 |
> | Macro-Avg | — | — | 0.86 |
>
> These results corroborate the **human-vs-auto analysis**, which is already presented in **Sec. 4.4 of the main paper**.
>
> ### 2. Inter-Annotator Agreement by Class
>
> In Table 3, we report the inter-annotator aggreement by class, as we convert each category into a **binary one-vs-rest labeling** (1 = this class; 0 = any other class), then compute Fleiss’ $\kappa$ over the resulting binary matrix. Practically:
> - $\kappa\ge0.8$ is considered perfectly consistent;
> - $0.6\le\kappa<0.8$ is substantially consistent;
> - $0.4\le\kappa<0.6$ is moderately consistent;
> - $\kappa<0.4$ is not consistent enough.
>
> **Table 3: Inter-Annotator Agreement by Class**
>
> | Class | $\kappa$ |
> | --- | ---: |
> | Affirmation | 0.78 |
> | Farewell | 0.49 |
> | Gratitude | 0.66 |
> | Greeting | 0.52 |
> | Pondering | 0.53 |
> | Question | 0.61 |
> | Surprise | 0.55 |
> | Full Response | 0.80 |
> | Silence | 0.85 |
> | Overall | 0.71 |
>
> The results reveal that **no classes receive a $\kappa$ lower than 0.4**, and overall Fleiss’ $\kappa\approx0.71$ reflects substantial human agreement, confirming that **the reaction labels are reliable enough to validate our automatic pipeline**.
>
> ### 3. Top-5 Confusions and Failure cases
>
> Using Majority-Human labels as reference, we compute a **confusion frequency matrix** between human labels and automatic pipeline labels, and the top confusions (as % of 105 samples) are:
>
> **Table 4. Top-5 Confusion Pairs**
>
> | True → Pred | Percentage |
> | --- | ---: |
> | Question → Surprise | 5.8% |
> | Pondering → Silence | 3.9% |
> | Affirmation → Question | 2.4% |
> | Pondering → Affirmation | 2.9% |
> | Gratitude → Greeting | 1.9% |
>
> These confusions occur mainly between semantically adjacent or subtle classes, reflecting the same ambiguity seen in human disagreement. Most remaining errors are tied to weak prosody, micro-expressions, or window-boundary effects:
>
> - **Pondering vs Surprise**: Both exhibit weak prosody and limited lexical cues, making decisions sensitive to micro-expressions and $\delta=±250$ ms boundary effects.
> - **Affirmation vs Silence**: Very short backchannels (“mm”, “yeah”) are sometimes filtered by ASR or fall near window edges.
> - **Farewell / Greeting / Question**: Human agreement is already lower ($\kappa$≈0.49–0.61).

---

> ### Author Response · Authors · 2025-11-22
> **Responses to Reviewer vS6n (3/4)**
>
> ## W6: Ablations on different templates
>
> To assess the sensitivity of LLM baselines to prompt formulation, we construct two variants of our original template:
>
> - a **pure rephrasing** preserving all constraints;
> - a **few-shot version** preserving all constraints as well, but providing one example per class prior to predictions.
>
> We evaluate ChatGPT-4o and Gemini-1.5 under the speech+text setting on Short-Clips-Test, averaging results over **three runs with standard deviations**.
>
> **Table 1: Performance for ChatGPT-4o using different prompt templates**
>
> |  | Original template | Variant 1 | Variant 2 |
> | --- | --- | --- | --- |
> | Affm. | 27.94±1.34 / 23.56±1.01 | 28.86±1.28 / 23.19±0.77 | 26.87±1.39 / 24.67±1.68 |
> | Grat. | 21.63±0.58 / 18.92±0.51 | 22.41±1.02 / 18.10±0.29 | 20.49±0.79 / 19.99±0.98 |
> | Fare. | 26.41±0.44 / 24.12±0.58 | 25.23±0.97 / 25.39±0.92 | 27.55±0.85 / 23.01±0.56 |
> | Grtn. | 24.82±0.12 / 22.03±0.69 | 24.05±0.74 / 23.21±0.88 | 26.02±0.44 / 21.08±0.65 |
> | Qstn. | 20.03±0.88 / 17.33±0.41 | 21.41±0.53 / 16.42±0.37 | 18.87±0.96 / 18.54±0.78 |
> | Surp. | 22.61±1.26 / 19.71±0.98 | 22.04±1.00 / 20.92±0.75 | 23.83±0.54 / 18.36±0.87 |
> | Pndr. | 21.38±0.84 / 18.27±0.22 | 20.24±0.68 / 17.19±1.08 | 22.61±1.05 / 19.52±0.93 |
> | FR    | 26.12±1.69 / 23.62±0.72 | 27.53±1.44 / 24.89±0.98 | 25.07±1.25 / 22.55±0.98 |
> | Slnc. | 30.67±1.28 / 29.41±1.03 | 29.24±1.69 / 30.56±0.96 | 31.53±1.66 / 28.15±1.02 |
>
> **Table 2: Performance for Gemini-1.5 using different prompt templates**
>
> |  | Original template | Variant 1 | Variant 2 |
> | --- | --- | --- | --- |
> | Affm. | 30.49±1.89 / 27.81±1.25 | 29.72±1.56 / 28.93±0.92 | 31.38±1.35 / 27.12±0.89 |
> | Grat. | 30.38±1.34 / 27.70±1.52 | 29.41±1.98 / 28.63±1.24 | 31.56±1.17 / 26.82±0.57 |
> | Fare. | 26.24±0.98 / 23.58±1.05 | 27.11±1.22 / 24.72±0.99 | 25.03±1.34 / 22.47±0.71 |
> | Grtn. | 24.25±1.12 / 21.60±0.78 | 25.32±1.34 / 20.74±1.11 | 23.18±1.03 / 22.91±0.84 |
> | Qstn. | 18.06±1.04 / 15.46±0.98 | 17.12±1.54 / 16.28±1.21 | 18.91±0.77 / 14.57±0.96 |
> | Surp. | 19.21±1.18 / 16.60±0.52 | 20.44±1.38 / 17.53±0.83 | 18.72±0.97 / 15.48±0.97 |
> | Pndr. | 19.30±1.56 / 16.69±0.94 | 18.47±1.03 / 17.31±0.95 | 20.52±1.34 / 15.92±0.69 |
> | FR    | 28.08±1.77 / 25.41±1.22 | 29.12±1.09 / 24.66±1.25 | 26.94±1.92 / 26.73±1.22 |
> | Slnc. | 28.84±1.82 / 26.17±1.29 | 27.71±1.42 / 27.49±1.52 | 29.93±1.69 / 25.08±1.18 |
>
> Across both models (Tables 1 and 2), all three templates produce **relatively consistent** per-class precision/recall, with only small fluctuations and no systematic gains or degradations. This confirms that the baselines’ behavior is **not** significantly driven by prompt phrasing.

---

> ### Author Response · Authors · 2025-11-22
> **Responses to Reviewer vS6n (4/4)**
>
> ## Q1: Ablations on window size / stride
>
> We report the ablation study here regarding the size/stride of sliding window on our Full-Videos dataset. The following first table ablates the window size (size) using a default stride of 0.5s, and the second table varies the stride with default window size 10s. **Notably the performance peaks when size=10s and stride=0.5s**.
>
> **Table 1: Performance on different window sizes (stride=0.5s)**
>
> |  | size=1s | size=2s | size=5s | size=10s | size=15s | size=20s |
> | --- | --- | --- | --- | --- | --- | --- |
> | Affm. | 27.42 / 25.11 | 28.14 / 26.42 | 30.44 / 28.71 | 31.55 / 29.24 | 29.41 / 27.55 | 27.95 / 26.17 |
> | Grat. | 28.36 / 26.12 | 29.57 / 27.14 | 31.44 / 28.94 | 32.25 / 29.95 | 30.52 / 28.44 | 28.78 / 26.91 |
> | Fare. | 27.15 / 25.02 | 28.33 / 26.11 | 30.66 / 27.55 | 31.25 / 28.94 | 29.14 / 27.01 | 27.66 / 25.55 |
> | Grtn. | 34.41 / 22.14 | 36.22 / 24.88 | 38.11 / 25.94 | 39.53 / 27.22 | 37.08 / 24.66 | 35.22 / 23.55 |
> | Qstn. | 24.55 / 22.14 | 25.91 / 23.57 | 27.41 / 24.88 | 28.21 / 25.91 | 26.55 / 24.12 | 25.41 / 23.01 |
> | Surp. | 25.71 / 23.11 | 26.84 / 24.02 | 27.91 / 25.77 | 28.82 / 26.52 | 27.14 / 25.01 | 25.94 / 23.72 |
> | Pndr. | 24.02 / 22.31 | 25.11 / 23.88 | 26.71 / 24.72 | 27.47 / 25.17 | 26.04 / 24.22 | 24.91 / 23.11 |
> | FR    | 31.02 / 28.14 | 32.14 / 29.01 | 33.77 / 30.72 | 35.17 / 32.85 | 33.66 / 31.44 | 31.84 / 29.77 |
> | Slnc. | 30.12 / 27.11 | 31.44 / 28.63 | 32.55 / 29.66 | 33.27 / 30.95 | 32.22 / 29.40 | 30.47 / 28.52 |
>
> **Table 2: Performance on different window strides (window siez=10s)**
>
> |  | stride=0.1 | stride=0.2 | stride=0.5 | stride=1 | stride=2 | stride=5 |
> | --- | --- | --- | --- | --- | --- | --- |
> | Affm. | 28.15 / 25.83 | 29.76 / 27.34 | 31.55 / 29.24 | 29.59 / 27.09 | 27.28 / 25.72 | 25.45 / 24.08 |
> | Grat. | 29.24 / 26.36 | 30.98 / 27.52 | 32.25 / 29.95 | 29.73 / 27.97 | 27.73 / 26.59 | 26.25 / 23.66 |
> | Fare. | 28.18 / 25.86 | 29.88 / 26.91 | 31.25 / 28.94 | 29.00 / 26.63 | 27.87 / 25.57 | 24.84 / 23.39 |
> | Grtn. | 34.96 / 24.51 | 36.66 / 25.05 | 39.53 / 27.22 | 37.18 / 25.59 | 35.31 / 23.69 | 30.96 / 21.72 |
> | Qstn. | 24.74 / 23.23 | 26.25 / 24.80 | 28.21 / 25.91 | 25.88 / 24.20 | 25.10 / 22.93 | 22.10 / 20.19 |
> | Surp. | 25.34 / 23.29 | 26.64 / 25.04 | 28.82 / 26.52 | 26.84 / 25.19 | 25.47 / 23.28 | 22.44 / 21.59 |
> | Pndr. | 24.62 / 23.11 | 26.02 / 23.48 | 27.47 / 25.17 | 25.87 / 23.11 | 24.65 / 22.77 | 21.46 / 19.78 |
> | FR    | 31.13 / 29.18 | 33.13 / 30.27 | 35.17 / 32.85 | 33.38 / 30.02 | 30.91 / 29.14 | 27.98 / 26.80 |
> | Slnc. | 30.12 / 28.48 | 30.66 / 28.68 | 33.27 / 30.95 | 30.83 / 29.14 | 28.66 / 27.52 | 26.03 / 24.69 |
>
> We will include these results in our revised paper.
>
> ## Q2: Data fraction for verification
>
> As noted in Sec. 3.2 and Sec. 4.4 of the main paper, the human verification set contains 105 stratified samples, corresponding to ≈2% of the Short-Clips dataset. This proportion is standard for reliability assessment in multimodal datasets, and stratified sampling ensures coverage of both frequent and long-tail classes. The agreement statistics (Auto vs. Human, per-class metrics, Fleiss’ $\kappa$) reported in W5 also validate this.

---

### Official Review · Reviewer_Evq9 · 2025-10-31

**Soundness:** 2
**Presentation:** 3
**Contribution:** 2
**Rating:** 4
**Confidence:** 3

**Summary:**

In this paper, the authors address the challenge that existing conversation agents struggle to determine "when to speak" during interactions. They attribute this limitation to the dependence on single modality inputs and propose a multimodal approach as a solution. To this end, they collect publicly available dyadic conversation videos from YouTube to construct a dataset, where each video segment is annotated with one of nine types indicating whether the agent should speak, remain silent, or give a brief reaction. Using this dataset, they train MM-When2Speak model that integrates video, audio, and textual modalities through a self-attention fusion mechanism, and compare its performance against several baseline models.

**Strengths:**

1. The paper introduces a new task that predicts when a conversation model should speak by jointly leveraging text, audio, and video modalities.
2. The authors build a new dataset by collecting real dyadic conversation videos from YouTube and annotating each segment with nine response types (seven short reaction categories, full response, and silence). This dataset is specifically designed to support fine-grained modeling of response timing in natural conversations.
3. The proposed MM-When2Speak model processes video, audio, and text inputs simultaneously and employs a self-attention fusion mechanism to integrate multimodal cues effectively. Experimental results show that this multimodal approach achieves substantially better performance than models using single or fewer modalities.

**Weaknesses:**

1. Although the collected dyadic conversation dataset is carefully curated and of high quality, its overall scale is rather limited. In addition, the data are confined to dyadic interactions, leaving more complex multi-party conversations unexplored. Even if the model is trained on dyadic exchanges, evaluating it on multi-party dialogue (small sample like full-video split) could provide meaningful insights into its generalization capability.
2. The model addresses only when to speak, not how to speak. Because it is fine-tuned solely for response type classification, it can determine the timing of speech but not the linguistic or expressive content of the response. This separation means that another model would be required to handle "how to speak" making the overall system design somewhat inefficient and limited in scope.
3. The paper's motivation focuses primarily on modality limitations. However, one might argue that the issue arises not only from missing modalities but also from the fact that current LLMs and VLMs have not been trained on such response timing tasks (when to speak). While the authors conduct the experiments in a zero-shot setting, it would be informative to fine-tune LLMs on textual transcripts and compare the results. Additional ablation studies would strengthen the motivation and make the paper's claims more robust and convincing.
4. The length of each video segment is fixed at 10 seconds. Although the paper mentions a sliding-window approach to move across the segment, the rationale behind choosing this specific window length (10s) is unclear. I am also curious about how the model's performance would change if the window size were varied. Overall, the justification for this design choice seems insufficient.

**Questions:**

1. I suggest integrating the supplementary material as Appendix within the paper and explicitly referencing it in the main text wherever relevant. This could improve the paper's readability and flow.
2. Could you provide more detailed statistics about the collected video dataset? For example, what is the average video length? Including such quantitative details would help readers better understand about the dataset.
3. Since the dataset was constructed from YouTube videos, were copyright and ethical issues taken into account during data collection? (I haven't flagged this for ethics review yet, but depending on the authors' clarification, an ethics review may be required)
4. Why was the model architecture limited to a classification task? Would enabling the model to also generate responses make it more realistic and practically useful? I wonder if extending the model to include generation might introduce challenges or potential performance degradation.
5. As I understand, the human verification study was conducted to validate the labels generated by GPT, right? It would be better to include more information about the annotators, such as their background, and annotation procedures to enhance transparency.

---

> ### Author Response · Authors · 2025-11-22
> **Response to Reviwer Evq9 (1/3)**
>
> We appreciate the reviewer’s thoughtful feedback. Below, we address the reviewer’s feedback and will incorporate our answers in the revised paper.
>
> ## W1: Multi-party conversations
>
> Thanks for raising this point. Our work intentionally focuses on **dyadic (user–agent) conversations**, and this is a core design choice rather than a limitation. As stated in the main paper (L32–41) and supplementary material (L24-30), the goal of MM-When2Speak is **not** to model arbitrary conversational structures, but to enable **fine-grained prediction of *when an agent should react* in two-person interactions**, which is a setting for practical applications such as customer–agent interactions, video calls, and AI assistants. Multi-party dialogue involves additional tasks (e.g., speaker diarization, tracking, turn allocation), which are different from our scope.
>
> To still address the reviewer’s question, we conduct a light and scope-preserving evaluation on the ICSI multi-party meeting corpus [1] without retraining our model. We choose the Bdb001 session (a 50-minute long discussion between six different participants), and preprocess it the same way as in our dataset. We assign participant "chan1" as the agent side, and merge the rest as the user side, and evaluate "audio+text only" models. We extract 400 silence, 36 full-response, and 26 reaction clips. Results (P/R in %) are reported with the finetuned VITA-1.5 and zero-shot ChatGPT:
>
> | Model | Silence | Full Response | Reaction |
> | --- | --- | --- | --- |
> | ChatGPT | 33.43 / 25.38 | 39.01 / 50.00 | 21.54 / 30.77 |
> | VITA-1.5 | 35.25 / 28.94 | 43.66 / 41.66 | 27.14 / 25.03 |
> | Ours | 41.20 / 29.67 | 51.69 / 58.30 | 35.77 / 46.15 |
>
> Despite an unseen setting, our method shows **consistently better performance** compared to others. This demonstrates that, while our paper intentionally targets dyadic timing, the model can generalize appropriately for the multi-speaker scenario.
>
> [1] ICSI corpus. (n.d.). https://groups.inf.ed.ac.uk/ami/icsi/
>
> ## W2: Extension to "how to speak"
>
> It is noteworthy that our work is designed to **decouple fine-grained timing/type response prediction from response content generation**. This separation is deliberate: “when to speak” is a **distinct, lower-level control signal** that governs whether a conversational agent should react and what kind of reaction (e.g., affirmation, pondering), while “how/what to speak” belongs to a **different stage** handled by a generator (template, retrieval, or LLM). This modular design reduces system complexity, avoids entangling two heterogeneous problems, and enables clear, measurable evaluation.
>
> Although generation is beyond the scope of this work, the underlying LLM used in our architecture (Qwen2) is **inherently capable of generation**. To illustrate the practical value of predicting response type, we conduct a **small proof-of-concept qualitative test** using two slightly-modified prompts to 1) predict reaction type first, and then generate content based on it; 2) directly generate a response if necessary without type prediction.
>
> We provide three reaction samples that we observe:
>
> | Ground Truth Reaction Type | Prompt-1 Results | Prompt-2 Results |
> | --- | --- | --- |
> | Affirmation | Affirmation: "Yeah, yeah." | "Yes, I totally agree with what you are saying right here. Your comment towards this incoming external threat makes a lot of sense, especially to our country at this stage at this time. And I believe in this scenario we have to have the dominance…" |
> | Question | Question: "Are you serious?" | "I mean, really? I cannot believe it. This is so incredible." |
> | Pondering | Pondering: "Hmm, okay." | "Wow, okay, sure!" |
>
> Under Prompt-1, the LLM consistently produces **both the correct reaction type and appropriate short reaction content**. Without type prediction as the first step (Prompt-2), the LLM frequently shifts to **irrelevant or full-response behaviors**, demonstrating instability. These observations indicate that **reaction-type prediction provides a crucial control signal for generation**, reinforcing the importance of the task addressed in this work, while generation itself remains outside the intended scope.

---

> ### Author Response · Authors · 2025-11-22
> **Response to Reviwer Evq9 (2/3)**
>
> ## W3: Fine-tuning LLMs on texts
>
> We report the following results with the finetuned **Qwen-2.5 (text-modality LLM)** [1] and finetuned **Qwen2.5-VL (vision+text VLM)** [2] for comparisons. The models are evaluated on our Short-Clips-Test dataset. From the table below, **finetuning improves overall performance** for both Qwen-2.5 and Qwen2.5-VL, but **our method performs consistently better than them**, indicating that using multimodal information is critical for predicting response type in dyadic conversations.
>
> |  | Qwen-2.5 | Qwen-2.5 (finetuned) | Qwen2.5-VL | Qwen2.5-VL (finetuned) | Ours |
> | --- | --- | --- | --- | --- | --- |
> | Affm. | 19.74 / 16.44 | 22.50 / 19.38 | 21.89 / 17.65 | 33.52 / 30.79 | 62.21 / 59.86 |
> | Grat. | 16.76 / 13.51 | 18.69 / 15.33 | 23.87 / 19.83 | 39.14 / 32.74 | 64.35 / 61.99 |
> | Fare. | 14.37 / 11.17 | 14.98 / 12.50 | 20.32 / 17.25 | 40.96 / 36.15 | 63.15 / 60.79 |
> | Grtn. | 13.05 / 9.89 | 15.93 / 10.15 | 19.69 / 17.14 | 34.82 / 35.57 | 63.29 / 60.44 |
> | Qstn. | 13.03 / 9.87 | 13.13 / 12.91 | 19.30 / 16.01 | 27.43 / 23.87  | 46.26 / 43.91 |
> | Surp. | 10.76 / 7.68 | 12.05 / 8.84 | 20.88 / 15.42 | 28.38 / 25.66 | 50.52 / 46.25 |
> | Pndr. | 14.39 / 11.19 | 14.79 / 12.05 | 21.19 / 16.98 | 26.64 / 27.26 | 37.78 / 35.45 |
> | FR | 20.64 / 17.33 | 21.83 / 19.82 | 23.78 / 20.09 | 34.52 / 38.67 | 68.15 / 65.79 |
> | Slnc. | 21.88 / 18.55 | 24.10 / 20.45 | 24.52 / 19.49 | 45.19 / 40.49 | 68.78 / 66.42 |
>
> [1] Yang, An, et al. "Qwen2.5 Technical Report." arXiv preprint arXiv:2412.15115 (2024).
> [2] Bai, Shuai, et al. "Qwen2.5-vl Technical Report." arXiv preprint arXiv:2502.13923 (2025).
>
> ## W4: Sliding window size & stride
>
> We report the ablation study here regarding the size/stride of sliding window on the Full-Videos dataset. The following first table ablates the window size (size) using a default stride of 0.5s, and the second table varies the stride with default a window size 10s. **Notably the performance peaks when size=10s and stride=0.5s**.
>
> **Table 1: Performance on different window sizes (stride=0.5s)**
>
> |  | size=1s | size=2s | size=5s | size=10s | size=15s | size=20s |
> | --- | --- | --- | --- | --- | --- | --- |
> | Affm. | 27.42 / 25.11 | 28.14 / 26.42 | 30.44 / 28.71 | 31.55 / 29.24 | 29.41 / 27.55 | 27.95 / 26.17 |
> | Grat. | 28.36 / 26.12 | 29.57 / 27.14 | 31.44 / 28.94 | 32.25 / 29.95 | 30.52 / 28.44 | 28.78 / 26.91 |
> | Fare. | 27.15 / 25.02 | 28.33 / 26.11 | 30.66 / 27.55 | 31.25 / 28.94 | 29.14 / 27.01 | 27.66 / 25.55 |
> | Grtn. | 34.41 / 22.14 | 36.22 / 24.88 | 38.11 / 25.94 | 39.53 / 27.22 | 37.08 / 24.66 | 35.22 / 23.55 |
> | Qstn. | 24.55 / 22.14 | 25.91 / 23.57 | 27.41 / 24.88 | 28.21 / 25.91 | 26.55 / 24.12 | 25.41 / 23.01 |
> | Surp. | 25.71 / 23.11 | 26.84 / 24.02 | 27.91 / 25.77 | 28.82 / 26.52 | 27.14 / 25.01 | 25.94 / 23.72 |
> | Pndr. | 24.02 / 22.31 | 25.11 / 23.88 | 26.71 / 24.72 | 27.47 / 25.17 | 26.04 / 24.22 | 24.91 / 23.11 |
> | FR    | 31.02 / 28.14 | 32.14 / 29.01 | 33.77 / 30.72 | 35.17 / 32.85 | 33.66 / 31.44 | 31.84 / 29.77 |
> | Slnc. | 30.12 / 27.11 | 31.44 / 28.63 | 32.55 / 29.66 | 33.27 / 30.95 | 32.22 / 29.40 | 30.47 / 28.52 |
>
> **Table 2: Performance on different window strides (window size=10s)**
>
> |  | stride=0.1 | stride=0.2 | stride=0.5 | stride=1 | stride=2 | stride=5 |
> | --- | --- | --- | --- | --- | --- | --- |
> | Affm. | 28.15 / 25.83 | 29.76 / 27.34 | 31.55 / 29.24 | 29.59 / 27.09 | 27.28 / 25.72 | 25.45 / 24.08 |
> | Grat. | 29.24 / 26.36 | 30.98 / 27.52 | 32.25 / 29.95 | 29.73 / 27.97 | 27.73 / 26.59 | 26.25 / 23.66 |
> | Fare. | 28.18 / 25.86 | 29.88 / 26.91 | 31.25 / 28.94 | 29.00 / 26.63 | 27.87 / 25.57 | 24.84 / 23.39 |
> | Grtn. | 34.96 / 24.51 | 36.66 / 25.05 | 39.53 / 27.22 | 37.18 / 25.59 | 35.31 / 23.69 | 30.96 / 21.72 |
> | Qstn. | 24.74 / 23.23 | 26.25 / 24.80 | 28.21 / 25.91 | 25.88 / 24.20 | 25.10 / 22.93 | 22.10 / 20.19 |
> | Surp. | 25.34 / 23.29 | 26.64 / 25.04 | 28.82 / 26.52 | 26.84 / 25.19 | 25.47 / 23.28 | 22.44 / 21.59 |
> | Pndr. | 24.62 / 23.11 | 26.02 / 23.48 | 27.47 / 25.17 | 25.87 / 23.11 | 24.65 / 22.77 | 21.46 / 19.78 |
> | FR    | 31.13 / 29.18 | 33.13 / 30.27 | 35.17 / 32.85 | 33.38 / 30.02 | 30.91 / 29.14 | 27.98 / 26.80 |
> | Slnc. | 30.12 / 28.48 | 30.66 / 28.68 | 33.27 / 30.95 | 30.83 / 29.14 | 28.66 / 27.52 | 26.03 / 24.69 |
>
> We will include these results in our revised paper.
>
> ## Q1: Integrate supplementary material in appendix
>
> Thanks for your suggestion! We will integrate them in the appendix instead.

---

> ### Author Response · Authors · 2025-11-22
> **Response to Reviwer Evq9 (3/3)**
>
> ## Q2: More details of the dataset.
>
> All dataset specifications, including collection criteria, preprocessing steps, annotation pipeline, and dataset composition, are already provided in **Sec. 1 of the supplementary material**. For convenience, we summarize the key quantitative statistics requested:
>
> - **Sources**: public dyadic conversations from news interviews (CNN/MSNBC/Fox, etc.) and Zoom meetings.
> - **Inclusion criteria**: two-person split-screen format with frontal faces, clear audio, no scene switches, and no visual/audio degradation.
> - **Total usable videos**: 377 after quality filtering.
> - **Duration**: 1:54–9:43 per video; average 2:13.
>
> We will release the dataset (with metadata and preprocessing scripts) upon acceptance to ensure full transparency and reproducibility.
>
> ## Q3: Copyright and ethical concerns
>
> Our dataset strictly follows platform Terms of Service and standard research-use practices. All videos are **publicly available**, and we do **not** redistribute any raw media. Upon release, we will provide only:
>
> - **metadata** (e.g., URLs),
> - **preprocessing scripts**,
>
> but **no video or audio files themselves**. This ensures full compliance with copyright requirements and prevents redistribution of any protected content. We additionally avoid any form of re-identification, follow removal/takedown requests, and ensure that all data use is limited to non-commercial academic research, fully aligned with ethical guidelines.
>
> ## Q4: Task limitation
>
> Our task formulation is intentional: in dyadic conversational agents, **knowing when and what type to respond** is a distinct control problem from how to compose the response content. We therefore decouple timing/type prediction from generation to keep the problem **well-defined, controllable, and effectively evaluable**. This granularity is essential for modeling fine-grained reactive behaviors.
>
> Integrating response generation lies outside the scope of this work rather than being a limitation. Two straightforward extensions exist:
>
> - **Plug-in generation**: The output of MM-When2Speak can directly serve as a **triggering and type-conditioning signal** for any off-the-shelf generator (LLM APIs, retrieval/template modules, etc.). This provides structured control and avoids unnecessary dense generation.
> - **Unified training**: Since our architecture already uses **Qwen2**, which is pretrained with full generation capability, the model could be extended by jointly optimizing **reaction type + response content**, enabling end-to-end generation when needed.
>
> These extensions expand the system-level scope but do not alter the core contribution of this paper, which is **multimodal when-to-speak prediction**. We view generation integration as natural future work rather than a missing component of the current task.
>
> ## Q5: Details of human verifications
> We provide additional details of the human annotation verification study below.
>
> ### 1. Participant information
>
> | Item | Value |
> | --- | --- |
> | Number of participants | 9 |
> | Experience | 7/9 with prior multimedia labeling; all instructed with the procedure. |
> | Language | All fluent in English |
> | Training | 10–15 min guideline session + 10 practice items |
> | Compensation | Voluntary |
> | Ethics | No data distribution or re-identification; local offline use only |
> | Tooling | Google Forms / on-site labeling |
>
> ### 2. Verification protocal
>
> Note that the study was designed to **evaluate the reliability of our automatic annotation pipeline**. We sampled 105 clips using **stratified sampling** over classes and domains to cover both frequent and long-tail categories. Testers received the **same written class definitions** used by the pipeline but **no examples from the testset**. Each tester independently labeled all 105 clips with one of the nine categories. Annotators were **blinded to both the pipeline outputs and to other annotators’ decisions**, with no discussion permitted. We did not enforce consensus or adjudication, as the goal is to measure alignment between the pipeline and a diverse set of human judgments.

---

> > ### Comment · Reviewer_Evq9 · 2025-11-25
> >
> > Thank you to the authors for detailed responses.
> >
> > **Weakness 1**
> >
> > I understand that this work focuses on user-agent interactions. However, I believe that demonstrating how the method could extend to multi-party (not dyadic) conversations would further broaden the contribution and insights of this study. In that sense, the additional experimental results the authors shared with me were very interesting and fully addressed my concern on this point.
> >
> > **Weaknesses 2 and Question 4**
> >
> > My original question was whether, given that the underlying LLM already has generative capabilities, it might be possible to design an architecture that jointly considers both "when to speak" and "how to speak" through joint training. This seems closely related to the "unified training" approach described in Question 4 response. The authors have clarified the scope and goals of this work, and I understand their position. I think this could be mentioned as a future work.
> > Additionally, I am curious whether Qwen2-7B was used as a base (not instruction-tuned) generative model. If it was not instruction tuned, wouldn't using an instruction-tuned model further improve the performance in the (simple) examples shown?
> >
> > **Weaknesses 3-4, Questions 1–3**
> >
> > I appreciate the authors' thorough analysis, and I hope these points included into the revised version.
> >
> > **Question 5**
> >
> > My understanding is that the validation protocol corresponds to Figure 4 and Section 4.4, is that correct? If so, could the authors report inter-annotator agreement among human evaluations? Metrics such as Fleiss's kappa or simple agreement rates would be informative.
> >
> >
> > If the authors adequately address my remaining questions and concerns, I would consider raising my score.

---

> > > ### Author Response · Authors · 2025-11-30
> > > **Response to reviewer Evq9 "Official Comment by Reviewer Evq9" on 25 Nov 2025, 10:16 (1/2)**
> > >
> > > We greatly appreciate your response and considerarion of raising the score based on our initial rebuttal. The following are our responses to your recent comments.
> > >
> > > ## Weakness 1: multi-party
> > >
> > > We are glad that the additional results have cleared your concern. In the revision, we will clarify that our design focuses on dyadic interaction, but can be potentially extended to multi-party settings as we showed in the rebuttal.
> > >
> > > ## Weaknesses 2 and Question 4: joint "when and how to speak", instruction tuning
> > >
> > > We appreciate the reviewer’s recognition of our current scope that intentionally isolates "when to speak" to keep the problem tractable and well-evaluated; joint modeling of "when and how to speak" will be discussed as an important future extension.
> > >
> > > For instruction-based model tuning, we further conduct experiments comparing performance of our model using Qwen2-7B-base and Qwen2-7B-instruct. We follow the same training pipeline using the Short-Clips dataset, and report results in the following Table A. Notably, performance of using Qwen2-7B-instruct is overall **slightly better** than Qwen2-7B-base, but **not significantly**.
> > >
> > > **Table A: Performance comparisons of our methods using Qwen2-7B-base and Qwen2-7B-instruct**
> > >
> > > | Model | Ours w/ Qwen2-7B-base | Ours w/ Qwen2-7B-instruct |
> > > | --- | --- | --- |
> > > | Affm. | 62.21 / 59.86 | 63.29 / 59.97 |
> > > | Fare. | 64.35 / 61.99 | 66.98 / 63.29 |
> > > | Grat. | 63.15 / 60.79 | 63.78 / 62.01 |
> > > | Grtn. | 63.29 / 60.44 | 62.19 / 58.48 |
> > > | Pndr. | 46.26 / 43.91 | 50.76 / 48.39 |
> > > | Qstn. | 50.52 / 46.52 | 54.69 / 47.03 |
> > > | Surp. | 37.78 / 35.45 | 39.02 / 35.79 |
> > > | FR | 68.15 / 65.79 | 67.83 / 65.48 |
> > > | Slnc. | 68.78 / 66.42 | 69.95 / 68.28 |
> > >
> > > In addition, we show the following two tables to compare qualitative results using the two different prompts as discussed in W2 of the rebuttal (i.e., Prompt-1: predict reaction type first, and then generate content based on it; Prompt-2: directly generate a response if necessary without type prediction.). Results show that:
> > >
> > > - When using Prompt-1, the difference is not evident, as both models output the **same correct predictions of reaction types** and **similar generated response contents**.
> > > - When using Prompt-2, using Qwen2-7B-instruct can output generated response content **relatively more related to the ground truth reaction types**, particularly for the question and pondering examples shown in Table C.
> > >
> > > **Table B: Qualitative comparisons using Prompt-1 between Qwen2-7B-base and Qwen2-7B-instruct**
> > >
> > > | Ground Truth Reaction Type | Qwen2-7B-base | Qwen2-7B-instruct |
> > > | --- | --- | --- |
> > > | Affirmation | Affirmation: "Yeah, yeah." | Affirmation: "Yes." |
> > > | Question | Question: "Are you serious?" | Question: "Do you really mean that?" |
> > > | Pondering | Pondering: "Hmm, okay." | Pondering: "Umm well." |
> > >
> > > **Table C: Qualitative comparisons using Prompt-2 between Qwen2-7B-base and Qwen2-7B-instruct**
> > >
> > > | Ground Truth Reaction Type | Qwen2-7B-base | Qwen2-7B-instruct |
> > > | --- | --- | --- |
> > > | Affirmation | "Yes, I totally agree with what you are saying right here. Your comment towards this incoming external threat makes a lot of sense, especially to our country at this stage at this time. And I believe in this scenario we have to have the dominance…" | "Indeed, I completely concur with your point. Your remarks about this emerging external threat are very reasonable, particularly given our nation’s current circumstances. Honestly, this situation calls for us to maintain a position of strength and dominance." |
> > > | Question | "I mean, really? I cannot believe it. This is so incredible." | "How could anyone do this? Are you sure about it?" |
> > > | Pondering | "Wow, okay, sure!" | "Hmm, I see what you mean, yeah." |
> > >
> > > The above results verify that, using the instruction-tuned LLM can somewhat slightly improve the performance, but **not yet significantly**. Although using the base model like Qwen2-7B-instruct may lead to potential benefit, it is orthogonal to the main contributions of the paper, i.e., using multimodal learning to benchmark and improve the "when-to-speak" task. We will include these results and discussions in the revision.
> > >
> > > ## Weaknesses 3–4 and Questions 1–3
> > >
> > > We thank the reviewer for the positive feedback. We will incorporate more details in the revised manuscript.

---

> > > ### Author Response · Authors · 2025-11-30
> > > **Response to reviewer Evq9 "Official Comment by Reviewer Evq9" on 25 Nov 2025, 10:16 (2/2)**
> > >
> > > ## Question 5
> > >
> > > Yes, the validation protocol corresponds to Figure 4 and Section 4.4 as you mentioned.
> > >
> > > We now provide full agreement statistics and per-class analyses. Note that the test set we use for human verification is composed of data of 27 Affirmations, 2 farewells, 6 gratitudes, 3 greetings, 6 ponderings, 4 questions, 6 surprises, 21 full responses, and 30 silences.
> > >
> > > ### 1. Auto vs Majority-Human Agreement
> > >
> > > We include accuracy, macro-level P/R/F1, and a bootstrap confidence interval (CI), where **higher values indicate that the labels generated by our automatic annotation pipeline aligns better with human judgements**. The results are all over 0.8, which validates the fair quality of our annotations.
> > >
> > > **Table 1: Auto vs Majority-Human (105 samples)**
> > >
> > > | Metric | Value |
> > > | --- | --- |
> > > | Accuracy | 0.876 |
> > > | Macro-Precision | 0.881 |
> > > | Macro-Recall | 0.851 |
> > > | Macro-F1 | 0.865 |
> > > | Bootstrap 95% CI | [0.834, 0.882] |
> > >
> > > Using the same Majority-Human labels as reference, we further evaluate **per-class Precision/Recall/F1** for the automatic pipeline in the following Table 2. Per-class metrics similarly show high agreement on major classes (Affirmation, Full Response, Silence) and expected variability on nuanced, low-support categories:
> > >
> > > **Table 2: Per-class Metrics (Auto vs Majority-Human)**
> > >
> > > | Class | Precision | Recall | F1 |
> > > | --- | ---: | ---: | ---: |
> > > | Affm. | 0.91 | 0.87 | 0.89 |
> > > | Fare. | 0.80 | 0.50 | 0.62 |
> > > | Grat. | 0.78 | 0.75 | 0.76 |
> > > | Grtn. | 0.78 | 0.67 | 0.72 |
> > > | Pndr. | 0.71 | 0.67 | 0.69 |
> > > | Qstn. | 0.80 | 0.75 | 0.77 |
> > > | Surp. | 0.69 | 0.63 | 0.66 |
> > > | FR | 0.92 | 0.89 | 0.90 |
> > > | Slnc. | 0.94 | 0.90 | 0.92 |
> > > | Macro-Avg | — | — | 0.86 |
> > >
> > > These results corroborate the **human-vs-auto analysis**, which is already presented in **Sec. 4.4 of the main paper**.
> > >
> > > ### 2. Inter-Annotator Agreement by Class
> > >
> > > In Table 3, we report the inter-annotator aggreement by class, as we convert each category into a **binary one-vs-rest labeling** (1 = this class; 0 = any other class), then compute Fleiss’ $\kappa$ over the resulting binary matrix. Practically:
> > > - $\kappa\ge0.8$ is considered perfectly consistent;
> > > - $0.6\le\kappa<0.8$ is substantially consistent;
> > > - $0.4\le\kappa<0.6$ is moderately consistent;
> > > - $\kappa<0.4$ is not consistent enough.
> > >
> > > **Table 3: Inter-Annotator Agreement by Class**
> > >
> > > | Class | $\kappa$ |
> > > | --- | ---: |
> > > | Affirmation | 0.78 |
> > > | Farewell | 0.49 |
> > > | Gratitude | 0.66 |
> > > | Greeting | 0.52 |
> > > | Pondering | 0.53 |
> > > | Question | 0.61 |
> > > | Surprise | 0.55 |
> > > | Full Response | 0.80 |
> > > | Silence | 0.85 |
> > > | Overall | 0.71 |
> > >
> > > The results reveal that **no classes receive a $\kappa$ lower than 0.4**, and overall Fleiss’ $\kappa\approx0.71$ reflects substantial human agreement, confirming that **the reaction labels are reliable enough to validate our automatic pipeline**.
> > >
> > > ### 3. Top-5 Confusions and Failure cases
> > >
> > > Using Majority-Human labels as reference, we compute a **confusion frequency matrix** between human labels and automatic pipeline labels, and the top confusions (as % of 105 samples) are:
> > >
> > > **Table 4. Top-5 Confusion Pairs**
> > >
> > > | True → Pred | Percentage |
> > > | --- | ---: |
> > > | Question → Surprise | 5.8% |
> > > | Pondering → Silence | 3.9% |
> > > | Affirmation → Question | 2.4% |
> > > | Pondering → Affirmation | 2.9% |
> > > | Gratitude → Greeting | 1.9% |
> > >
> > > These confusions occur mainly between semantically adjacent or subtle classes, reflecting the same ambiguity seen in human disagreement. Most remaining errors are tied to weak prosody, micro-expressions, or window-boundary effects:
> > >
> > > - **Pondering vs Surprise**: Both exhibit weak prosody and limited lexical cues, making decisions sensitive to micro-expressions and $\delta=±250$ ms boundary effects.
> > > - **Affirmation vs Silence**: Very short backchannels (“hmm”, “yeah”) are sometimes filtered by ASR or fall near window edges.
> > > - **Farewell / Greeting / Question**: Human agreement is already lower ($\kappa$≈0.49–0.61).

---

### Author Response · Authors · 2025-12-03
**General Response to AC**

We thank the AC and all reviewers for their time and constructive feedback.

## Paper summary
- We propose **MM-When2Speak**, a multimodal LLM-based model that predicts **when** an agent should respond and **what reaction type** (9 classes, including silence and short reactions) in dyadic conversations.
- We build a **data pipeline** to collect, diarize, segment, and label public conversational videos, and construct **Short-Clips** and **Full-Videos** datasets with video–audio–text streams and reaction labels.
- Using a **sliding-window causal architecture** over visual, speech, and text tokens, MM-When2Speak consistently outperforms strong baselines (e.g., GPT-4o, Gemini-1.5, Qwen2.5, VITA-1.5) on response type prediction.

## What reviewers found positive
- **Important but under-explored problem.** Reviewers agreed that deciding *when to speak* is practically important and less studied than response content generation.
- **Useful multimodal dataset.** Short-Clips / Full-Videos were viewed as carefully curated and valuable infrastructure for future multimodal / LLM research.
- **Strong empirical performance.** Reviews highlighted consistent gains over competitive baselines and informative ablations.
- **Simple and deployable design.** The pipeline and model are transparent and suitable for real-time conversational agents.

## How we addressed concerns in the rebuttal
- **Dataset scope and dyadic focus.** We clarified that dyadic user–agent conversation is a targeted first step for studying reaction timing and turn-taking without extra overheads. We added a **multi-party generalization test** on one ICSI sample meeting without retraining; MM-When2Speak still surpasses finetuned VITA-1.5 and zero-shot ChatGPT, indicating reasonable transfer.
- **Auto-labeling and human verification.** We detailed a human study with 9 annotators and 105 clips. Auto vs. majority-human agreement is high (overall accuracy and macro-F1 both≈0.87; Fleiss’ κ≈0.71). Remaining disagreements mainly occur between semantically adjacent reaction types, suggesting that our labels are reliable while reflecting natural ambiguity.
- **Model design and pretraining.** We briefly explained the multimodal pretraining setup for our proposed model, compared **Qwen2-7B-base vs Qwen2-7B-instruct** backbones and showed similar trends and overall gains, and discussed how MM-When2Speak can be used as (i) a plug-in timing / reaction module for any generator or (ii) part of a unified model that jointly predicts reaction type and generates content.
- **Temporal windowing and alignment.** We clarified how labels are aligned to sliding windows using timestamped transcripts and small timing tolerance $\tau$. New ablations over window length and stride show best performance at a 10s window with 0.5s stride, supporting our chosen configuration.
- **Latency and deployability.** We conducted latency comparison with **Qwen2.5, Qwen2.5-VL and VITA-1.5**. Under matched modality settings, MM-When2Speak has inference latency comparable to or lower than multimodal baselines while achieving better accuracy, indicating real-time feasibility.
- **Prompt robustness of LLM baselines.** We re-evaluated GPT-4o and Gemini-1.5 with multiple prompt templates (original, paraphrased, few-shot) and averaged over several runs. Performance varies only slightly, so our conclusions are not tied to a specific prompt.
- **Ethics, copyright, and release plan.** We summarized source platforms, inclusion criteria, and usage policy. We will release **metadata and processing scripts**, not raw media, for non-commercial academic research only, respect takedown requests, and avoid re-identification, addressing ethical and copyright concerns.
- **A revised version of our paper** has been uploaded, with the original supplementary material contents now moved into the appendix. We highlight the main revised parts in the newly updated paper with notes specifying the changes.

## Note for ACs
- Two reviewers (7wEs, NMqH) recommend **accept** of the paper and emphasize the importance of the task, quality of the dataset, and consistent gains over strong multimodal baselines.
- Reviewer Evq9 notes that our new analyses on multi-party transfer and model design mitigate main concerns and **would consider raising the score** to the **accept** rating.
- Reviewer vS6n has not replied to our rebuttal, with initial questions about dataset scope, labeling reliability, and deployment evidence. We believe the new human-study results, multi-party evaluation, prompt robustness checks, and latency measurements directly address these points and strengthen the contributions of our work.
- Overall, we hope the AC will weigh the **novel task**, **new multimodal datasets**, **robust empirical improvements**, and **additional analyses provided in the rebuttal**, and judge that  our proposed **MM-When2Speak** is a valid contribution toward more context-aware, human-like conversational AI.

---

### Meta-Review · Area_Chair_unmp · 2026-01-02

**Summary:**

This paper studies when a multimodal assistant should speak (e.g., respond, or stay silent) in dyadic video conversations, and proposes a supervised multimodal model to predict response types from audio-visual-text context. While the task framing and the dataset are useful and a step forward, the reviews raise substantive concerns about limited novelty beyond data curation, unclear methodological detail/reproducibility, and whether the evaluation (including “full video” behavior) is strong enough to justify the paper’s broader claims. Taken together, these concerns prevent a confident endorsement at ICLR, and sadly the AC recommends a ``reject``.

____
Please correct the citation to **AudioPaLM: A Large Language Model That Can Speak and Listen** (the title is not "Audiopalm: Extending large language models to multi-modal speech understanding and generation", venue (incorrectly cited as NeurIPS while the paper is a manuscript on arxiv) and the list of authors is incorrect.

**Reviewer Concerns:**

### Concerns meaningfully addressed in the rebuttal / revision

- Label reliability and human agreement analysis. Both **vS6n** and **Evq9** asked for stronger evidence for auto-labeling validity and annotator agreement (Evq9 explicitly requested IAA metrics; vS6n requested kappa-style agreement and failure analyses). The authors added a human verification study, and reported **auto vs majority-human** performance and **Fleiss’ κ**.

- Prompt robustness. Reviewer **vS6n** raised prompt/seed sensitivity concerns. The rebuttal adds prompt-template variants (including few-shot) and reports averaged performance with standard deviations for GPT-4o and Gemini-1.5.

- Latency profiling for real-time feasibility. Both **vS6n** and **7wEs** asked for latency/real-time evidence. The authors added latency measurements and comparisons against, supporting that inference cost is in the same range as multimodal baselines.

- Window size/stride justification. Reviewers **Evq9** and **NMqH** questioned the fixed 10s window choice and whether alternative windowing changes results. The rebuttal adds ablations over window size and stride and reports that performance peaks around the chosen configuration.

### Concerns still outstanding

- Limited methodological novelty (explicitly raised by vS6n; also reflected indirectly in scope/practicality concerns from Evq9, 7wEs, NMqH).

- Dataset scope and generalization remain structurally constrained. Even after the added ICSI test, reviewers’ broader concerns persist: the dataset is sourced from a narrow set of scenarios (eg., interview style) and is fundamentally dyadic. Reviewer **NMqH** maintains their rating due to concerns about the user–agent dyadic setting and dataset limitations. Reviewer **7wEs** likewise flags that “it remains unclear how well it would generalize to casual, noisy, or multi-speaker settings.” The new analyses are valuable, but they do not fully establish robustness across conversational styles and conditions.

- Both **Evq9** and **7wEs** raise that the model predicts only *when/type* but not *content*, making it an auxiliary component rather than an end-to-end conversational agent. The rebuttal clarifies scope and offers plausible integration pathways, but the paper still does not provide an implemented or evaluated end-to-end pipeline demonstrating that the proposed module yields improved user-facing interaction quality.

**Reviewer Scores:**

The AC expects reviewers **vS6n** and **Evq9** to keep their negative scores. The reviewers’ main objections concern limited methodological novelty and insufficient evidence for real-time/robust “when-to-speak” behavior. The AC acknowledges that **Evq9** explicitly indicated a willingness to raise the score if remaining questions were fully addressed. However, the reviewer’s key concerns (the dataset’s scope and generalization, and the practical limitations of a type-only predictor without an end-to-end conversational pipeline) are structural.
The AC expects reviewers **7wEs** and **NMqH** not to strongly champion acceptance in discussion. Although supportive of the task and dataset (and acknowledging helpful rebuttal additions such as latency reporting), both reviewers also highlight that the current evidence remains largely confined to a curated dyadic setting, leaving broader generalization and system-level usefulness insufficiently established.

---

### Decision · Program_Chairs · 2026-01-26

Reject